# Germ layer-specific regulation of cell polarity and adhesion gives insight into the evolution of mesoderm

Miguel Salinas-Saavedra[1,2]*, Amber Q Rock[1], Mark Q Martindale[1,2]*

[1]The Whitney Laboratory for Marine Bioscience, University of Florida, Florida, United States; [2]Department of Biology, University of Florida, Florida, United States

**Abstract** In triploblastic animals, Par-proteins regulate cell-polarity and adherens junctions of both ectodermal and endodermal epithelia. But, in embryos of the diploblastic cnidarian *Nematostella vectensis*, Par-proteins are degraded in all cells in the bifunctional gastrodermal epithelium. Using immunohistochemistry, CRISPR/Cas9 mutagenesis, and mRNA overexpression, we describe the functional association between Par-proteins, ß-catenin, and *snail* transcription factor genes in *N. vectensis* embryos. We demonstrate that the aPKC/Par complex regulates the localization of ß-catenin in the ectoderm by stabilizing its role in cell-adhesion, and that endomesodermal epithelial cells are organized by a different cell-adhesion system than overlying ectoderm. We also show that ectopic expression of *snail* genes, which are expressed in mesodermal derivatives in bilaterians, is sufficient to downregulate Par-proteins and translocate ß-catenin from the junctions to the cytoplasm in ectodermal cells. These data provide molecular insight into the evolution of epithelial structure and distinct cell behaviors in metazoan embryos.
DOI: https://doi.org/10.7554/eLife.36740.001

**\*For correspondence:**
mssaavedra@whitney.ufl.edu (MS-S);
mqmartin@whitney.ufl.edu (MQM)

**Competing interests:** The authors declare that no competing interests exist.

## Introduction

Bilaterian animals comprise more than the 95% of the extant animals on earth and exhibit enormous body plan diversity (*Martindale and Lee, 2013*). One of the most important morphological features in bilaterian evolution is the emergence of the mesoderm, an embryological tissue that gives rise important cell types such as muscle, blood, cartilage, bone, and kidneys in the space between ecto-derm and endoderm. The emergence of mesoderm clearly contributed to the explosion of biological diversity throughout evolution (*Martindale and Lee, 2013*; *Martindale, 2005*). Cnidarians (e.g. sea anemones, corals, hydroids, and 'jellyfish') are the sister group to bilaterians, and despite their sur-prisingly complex genomes (*Putnam et al., 2007*), do not possess a distinct mesodermal tissue layer. Instead, the gastrodermal lining to their gut cavity consists of a bifunctional endomesodermal epithelium with molecular characteristics of both bilaterian endodermal and myoepithelial meso-dermal cells (*Martindale and Lee, 2013*; *Technau and Scholz, 2003*; *Martindale et al., 2004*; *Jahnel et al., 2014*). For example, *brachyury* and *snail*, among other genes, contribute to the specifi-cation of the endomesodermal fates in both bilaterian and cnidarian embryos (*Technau and Scholz, 2003*; *Martindale et al., 2004*; *Magie et al., 2007*; *Yasuoka et al., 2016*; *Servetnick et al., 2017*). Yet in bilaterians, mesodermal cells segregate from an embryonic endomesodermal precursor to form both endoderm and a third tissue layer (mesoderm) not present in the embryos of diploblastic cnidarians (*Martindale et al., 2004*; *Rodaway and Patient, 2001*; *Davidson et al., 2002*; *Maduro and Rothman, 2002*; *Solnica-Krezel and Sepich, 2012*). How mesodermal cells originally segregated from an ancestral endomesodermal epithelium during animal evolution is still unclear (*Martindale and Lee, 2013*; *Martindale, 2005*; *Technau and Scholz, 2003*), particularly because vir-tually all of the genes required for mesoderm formation are present in cnidarian genomes

**eLife digest** Most animals – including birds, fish and mammals – have symmetrical left and right sides, and are known as bilaterians. During early life, the embryos of animals in this group develop three distinct layers of cells: the ectoderm (outer layer), the endoderm (inner layer), and the mesoderm (middle layer). These layers then go on to form the animal's tissues and organs. The ectoderm produces external tissues, such as the skin and the nervous system; the endoderm forms internal tissues, like the gut; and the mesoderm creates all tissues in between, like muscles and blood.

Another, smaller group of animals, called cnidarians, do not have left and right sides. Instead, they have a 'radial symmetry', meaning they have multiple identical parts arranged in a circle. These animals – which include corals, jellyfish and sea anemones – only develop two distinct layers of cells, equivalent to the outer and inner layers of bilaterians. Cnidarians evolved before bilaterians, but their genetic material is equally complex. So why did these two groups evolve to have different layers of cells? And how exactly do animal embryos develop these distinct layers?

To address these questions, Salinas-Saavedra et al. studied embryos of the sea anemone *Nematostella vectensis*. Molecules called Par-proteins play an important role in controlling how cells behave and attach to one another (and therefore how they form layers). So, using a technique called immunohistochemistry to look inside cells, Salinas-Saavedra et al. examined these proteins in the two layers of cells in sea anemone embryos.

The experiments found that in the sea anemones, Par-proteins are arranged differently in cells that form the 'skin' compared to cells that form the 'gut'. In other words, cells in the outer layer attach to one another in a different way than cells in the inner layer, where the Par-proteins are degraded by 'mesodermal' genes. The findings also show that these sea anemones have all they need to form a third middle layer of cells. Like bilaterians, they could potentially move cells in and out of sheets that line surfaces inside the body – but they do not naturally do this.

Understanding how animals form different layers of cells is important for scientists studying evolution and the development of embryos. However, it also has wider applications. For instance, some cells involved in developing the mesoderm are also involved in forming tumors. Future research in this area could help scientists learn more about how cancer-like cells form in animals.

DOI: https://doi.org/10.7554/eLife.36740.002

(*Putnam et al., 2007*; *Baumgarten et al., 2015*; *Chapman et al., 2010*; *Shinzato et al., 2011*). During the last decade, several studies have described molecular and cellular characteristics related to the segregation of mesoderm during bilaterian development (*Solnica-Krezel and Sepich, 2012*; *Keller et al., 2003*; *Darras et al., 2011*; *Schäfer et al., 2014*). Here, we investigate the cellular basis of morphogenesis during embryogenesis of the 'diploblastic' sea anemone, *Nematostella vectensis*.

In most bilaterian embryos described to date, after a series of synchronous and stereotyped cleavage divisions, maternal determinants induce the localization of nuclear ß-catenin to blastomeres derived from the vegetal pole (*Martindale and Lee, 2013*). Hence, gastrulation and the specification of endomesodermal fates is restricted to the vegetal pole. In these species, *brachyury* is expressed at the border of the blastopore and *snail* is expressed in the prospective mesodermal tissues (*Technau and Scholz, 2003*). The formation of mesoderm involves a variety of cellular processes including the downregulation of E-cadherin, loss of apicobasal cell polarity, and in some cases, the induction of epithelial-to-mesenchymal transition (EMT) (*Solnica-Krezel and Sepich, 2012*; *Schäfer et al., 2014*; *Acloque et al., 2009*; *Lim and Thiery, 2012*).

Embryos of the cnidarian starlet sea anemone *N. vectensis* develop without a stereotyped cleavage pattern but cell fates become organized along the embryonic animal-vegetal axis (*Fritzenwanker et al., 2007*; *Salinas-Saavedra et al., 2015*). During blastula formation, embryonic cells of *N. vectensis* form a single hollow epithelial layer. Epithelial cells of the animal pole, characterized by the nuclear localization of Nvß-catenin prior to gastrulation (*Wikramanayake et al., 2003*; *Lee et al., 2007*), invaginate by apical constriction to form the endomesodermal epithelium (*Magie et al., 2007*; *Tamulonis et al., 2011*). The expression of *Nvbrachyury* around the presumptive border of the blastopore and *Nvsnail* genes in the presumptive endomesodermal gastrodermis

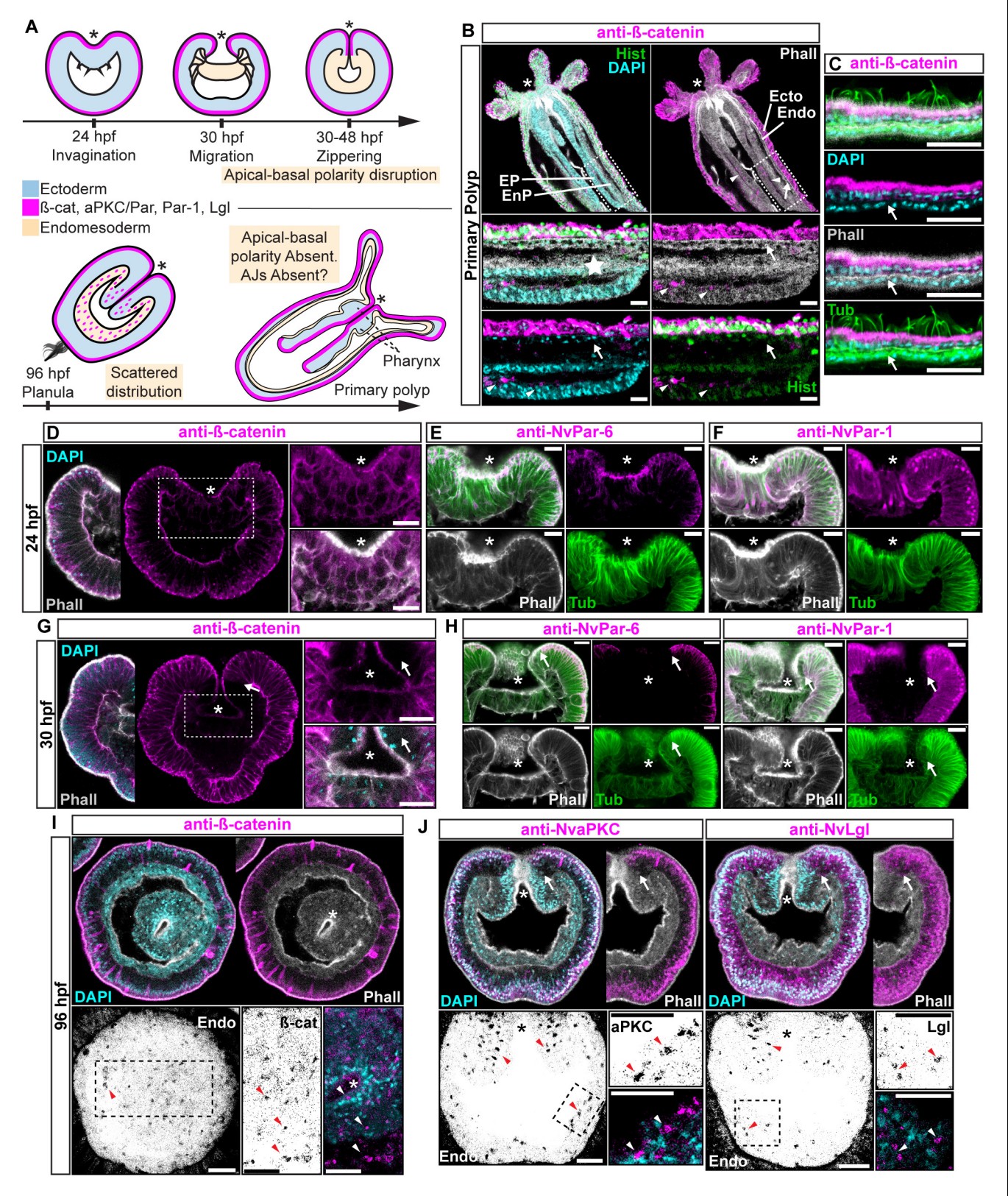

**Figure 1.** Components of the Par system and ß-catenin are downregulated from the *N. vectensis* endomesoderm during gastrulation. (A–F) Confocal images of immunofluorescent staining (IFS) of lateral views of gastrulation embryos (animal pole up). The * marks the site of gastrulation in all cases. Samples are counterstained with Phalloidin (Phall) staining (white) to show cell boundaries, DAPI to visualize cell nuclei (blue), and Tubulin antibody

*Figure 1 continued on next page*

*Figure 1 continued*

(Tub) staining is shown as counterstain (green). All images are a single optical section from a z-stack confocal series. All scale bars, 20 µm. (**A**) Summary diagram depicting the localization of ß-catenin and Par proteins at the observed stages. Pale boxes denote changes observed in the endomesoderm. (**B**) IFS for ß-catenin (magenta) in primary polyps. High magnification images from boxed region (endomesoderm, Endo) are shown on the bottom. Arrows indicate the absence of ß-catenin expression in the endomesoderm. Arrowheads indicate the ß-catenin expression in the ectodermal pharynx (EP). Star indicates the endomesodermal pharynx (EnP). Histone antibody (Hist) staining is shown as counterstain to show the penetrability in the fixed tissue. See also *Figure 1—figure supplement 1*. (**C**) IFS for ß-catenin (magenta) in the ecto and endomesoderm (arrow) of primary polyps. (**D**) IFS for ß-catenin (magenta) at 24 hpf shows localization to the apical domain where adherens junctions reside in all cells of the blastula. High magnification images from boxed region (prospective endomesoderm) are shown on the right. (**E**) IFS for *Nv*Par-6 (magenta) at 24 hpf showing the same sub-cellular localization as ß-catenin (**A**). High magnification images from boxed region in (**A**) (prospective endomesoderm) are shown on the right. Merged image shown on upper left. (**F**) IFS for *Nv*Par-1 at 24 hpf shows a complementary basolateral expression. High magnification images from boxed region (prospective endomesoderm) are shown on the right. (**G**) IFS for ß-catenin at 30 hpf shows the loss of expression of ß-catenin (magenta) in invaginating endomesoderm (box). The arrow (**D–F**) marks the boundary between ectoderm and invaginating endomesoderm. High magnification images from boxed region (prospective endomesoderm) are shown on the right. (**H**) IFS for *Nv*Par-6 and *Nv*Par-1(magenta) at 30 hpf show that all Par proteins are down regulated at the site of gastrulation. IFS for *Nv*Par-6 shows an even earlier down regulation than ß-catenin (**D**). High magnification images from boxed region (prospective endomesoderm) are shown on the right. Merged image shown on upper left. (**I**) Oral view of IFS for ß-catenin (magenta) at 96 hpf showing apical localization in overlying ectoderm, but absence in endomesodermal tissues. The two bottom panels show high magnifications of the endomesoderm region (image inverted). Arrowheads indicate the localization of ß-catenin expression (black) in some scattered endomesodermal cells. (**J**) Lateral view of IFS for *Nv*aPKC and *Nv*Lgl (magenta) at 96 hpf showing loss of expression in invaginating epithelial cells. The four bottom panels show high magnifications of the endomesoderm region (image inverted). Arrowheads indicate the localization of *Nv*aPKC and *Nv*Lgl proteins (black) in some scattered endomesodermal cells.

DOI: https://doi.org/10.7554/eLife.36740.003

The following figure supplement is available for figure 1:

**Figure supplement 1.** Epidermal and gastrodermal cells are joined by different set of junctional complexes.

DOI: https://doi.org/10.7554/eLife.36740.004

of *N. vectensis* embryos occurs even before the morphological process of gastrulation begins (*Scholz and Technau, 2003*; *Röttinger et al., 2012*).

Interestingly, the components of the intracellular polarity Par system (*Nv*aPKC, *Nv*Par-6, *Nv*Par-3, *Nv*Par-1, and *Nv*Lgl), which show a highly polarized bilaterian-like subcellular distribution throughout all epithelial cells at the blastula stage in *N. vectensis* (*Salinas-Saavedra et al., 2015*), are specifically degraded and down-regulated from the endomesoderm during the gastrulation process (*Figure 1A*). We have previously suggested that the expression of bilaterian 'mesodermal genes' (e.g. *Nvsnail*) might induce the loss of apicobasal cell-polarity indicated by the absence of the components of the Par system in the endomesoderm of *N. vectensis* embryos (*Salinas-Saavedra et al., 2015*). Recent studies in *N. vectensis* and bilaterians have provided information that supports this hypothesis. For example, it has been shown that *snail* is necessary and sufficient to downregulate Par3 in *Drosophila* mesoderm, inducing the disassembly of junctional complexes in these tissues (*Weng and Wieschaus, 2016*, *2017*). In addition, we have shown that *Nvbrachyury* regulates epithelial apicobasal polarity of *N. vectensis* embryos, suggesting some aspects of epithelial cell polarity are highly conserved (*Servetnick et al., 2017*). Together, this evidence suggests a plausible cellular and molecular mechanism for the segregation of a distinct cell layer in bilaterian evolution from an ancestral bifunctional endomesodermal tissue. Thus, in this study, we describe the functional association between the components of the Par system, apical junctions, epithelial integrity, and the nuclearization of *Nv*ß-catenin in a cnidarian embryo. In addition, we demonstrate that the endomesoderm in *N. vectensis* is organized by different junctional complexes that confer different functional properties to this tissue than the overlying ectoderm. And finally, we investigate the putative interactions between the components of the Par system, the canonical Wnt signaling pathway, and *snail* gene expression, giving insights on the evolution of the mesoderm and EMT.

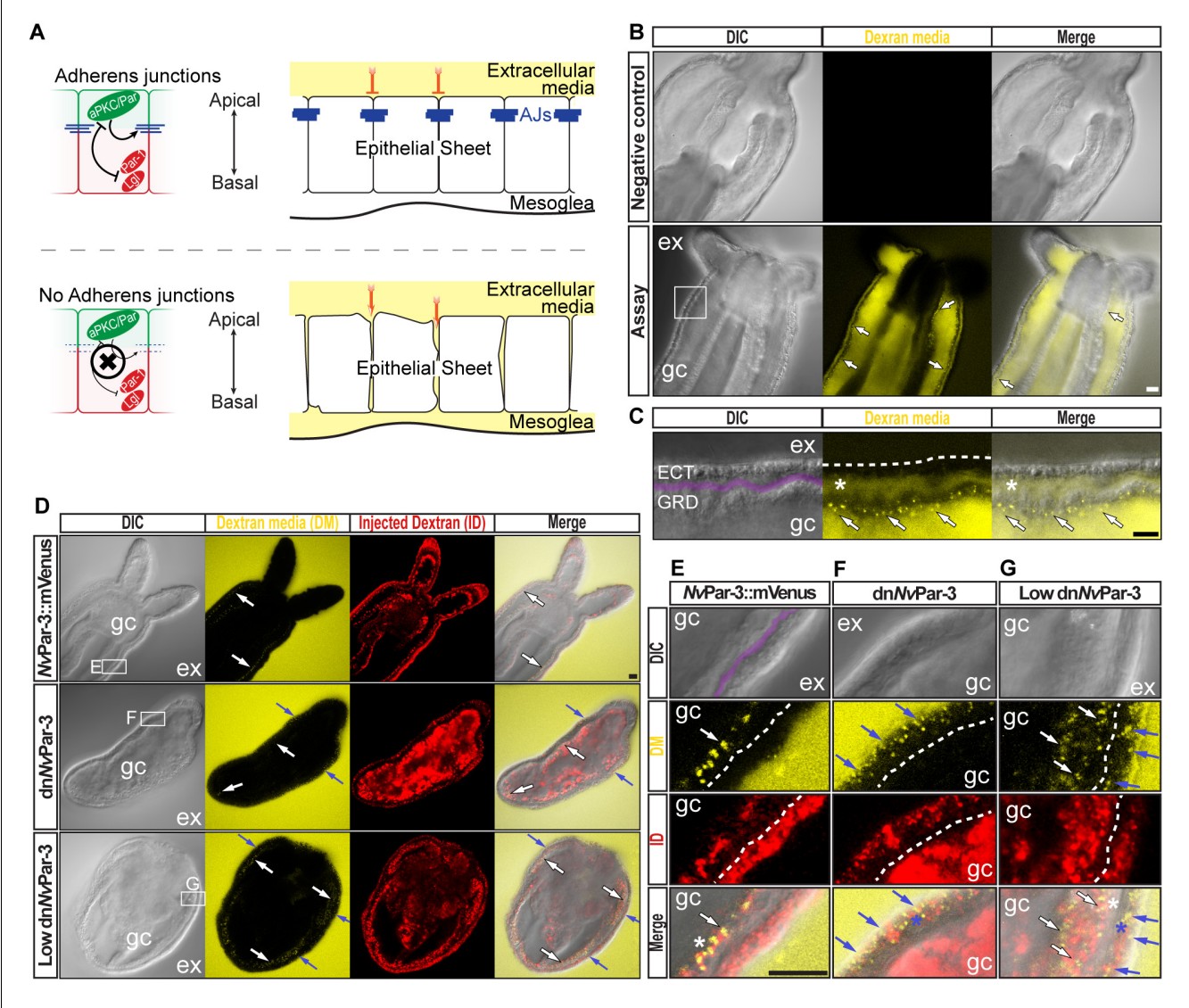

**Figure 2.** The aPKC/Par complex maintains Adherens Junctions (AJs) of ectodermal epithelial cells. Arrows indicate the direction of the flow: from gastric cavity (gc) to mesoglea (white) and from external media (ex) to ectoderm (blue). Dashed lines indicate the base of the epidermis. All images are single optical section from the z-stack confocal series. Scale bars, 20 μm. (A) Diagram depicting the hypothesis that when the aPKC/Par complex is functional (top row), AJs are present (blue stripes) and a paracellular epithelial barrier is formed. When aPKC/Par complex is not functional (bottom row), AJs are disrupted, the epithelial barrier is perturbed, and the extracellular solution moves paracellularly into the mesoglea. (B) Penetration assay of wild type (uninjected) primary polyps at low magnification showing the movement of 10,000 MW fluorescent dextran. Top row, no dextran. Bottom row, dextran (yellow) in the gc moves in to the mesoglea through paracellular spaces between gastrodermal cells (arrows). (C) High magnification images from box shown in (B). *: mesoglea (purple band) that separates the ectoderm (ECT, dashed line) from gastrodermis (GDR). Note the dye moving between cells from the gc media (arrows). (D) Low magnification images comparing polyps expressing *Nv*Par-3::mVenus and a dominant negative version of *Nv*Par-3 (dn*Nv*Par-3::mVenus) expressing-embryos. Dextran media (DM; extracellular) is pseudo-colored yellow. Dextran (red) was co-injected with mRNAs to label the cells and differentiate intracellular regions. mVenus channel was omitted for better visualization (shown in *Figure 2—figure supplement 1*). Lower concentrations of dn*Nv*Par-3 were injected to preserve endomesodermal tissues. Note that the dextran media was found between the cells labeled in red. See also *Figure 3—figure supplement 1* for dn*Nv*Par-3 description. (E) High magnification images from (E) boxed region in (D). Purple band depicts Mesoglea. (F) High magnification images from (F) boxed region in (D). (G) High magnification images from (G) boxed region in (D). *: Paracellular spaces of both, the epidermis (blue) and gastrodermis (white).

DOI: https://doi.org/10.7554/eLife.36740.005

The following figure supplement is available for figure 2:

**Figure supplement 1.** mVenus protein expression related to *Figure 2*.
DOI: https://doi.org/10.7554/eLife.36740.006

# Results

## Ectodermal and endomesodermal epithelia are organized by different cell-cell adhesion complexes

Components of the Par system are not present in the cells of endomesodermal epithelium of *N. vectensis* during gastrulation, even though the very same cells express these proteins during the blastula stage (*Salinas-Saavedra et al., 2015*) (*Figure 1*). This absence is consistent with the absence of apical Adherens Junctions (AJs) in the endomesoderm of *N. vectensis* (*Figure 1—figure supplement 1*) and other cnidarians (*Magie et al., 2007*; *Chapman et al., 2010*; *Ganot et al., 2015*). At polyp stages, neither ß-catenin (an AJ-associated protein) (*Figure 1B and C*) nor the Par proteins (*Figure 1—figure supplement 1C*) are detectable in endomesodermal cells of either the gastrodermis or the pharynx. When *N. vectensis* embryos are stained with antibodies to ß-catenin (*Figure 1*) or if *Nv*ß-catenin::GFP mRNA is expressed in uncleaved zygotes (*Figure 1—figure supplement 1B*), clear localization of ß-catenin can be seen in the cortex of ectodermally derived epithelial cells (*Figures 1B, C, D, G and I*), but not in endomesodermal cells (*Figure 1B and C*). In pharyngeal cells that are located between the epidermis and gastrodermis, *Nv*ß-catenin (*Figure 1B and D*), *Nv*Par-6 (*Figure 1E*), and *Nv*Par-1 (*Figure 1F*) expression begins to disappear, and is localized only in the most apical regions, indicating that AJs are being disassembled/degraded during the gastrulation process (*Figure 1G and H*). During later planula stages, ß-catenin and the components of the Par system display scattered patterns in the cytoplasm of a small subset of endomesodermal cells (*Figure 1I and J*). Even though we do not know the identity of these cells, this expression temporally coincides with the transient activation of Wnt signaling emanating from the oral pole (*Kusserow et al., 2005*; *Marlow et al., 2013*) at those developmental stages. In bilaterians (*Acloque et al., 2009*; *Lim and Thiery, 2012*) and *N. vectensis* (*Kusserow et al., 2005*; *Marlow et al., 2013*), the later activation of Wnt signaling is also associated with neurogenesis, and may cause the observed changes in protein localization.

Regardless of this scattered expression, it is clear that cells that undergo gastrulation in *N. vectensis* lose their polarized ectodermal cell-cell adhesion complex and components of the Par system, including ß-catenin, are downregulated from endomesodermal tissues (*Figure 1*). In bilaterians, the proper formation of an epithelial paracellular barrier (essential for tissue homeostasis) depends on the establishment of adhesive complexes between adjacent cells (*Higashi et al., 2016*; *Jonusaite et al., 2016*), which are regulated by the aPKC/Par complex (*Ohno et al., 2015*). To test if this absence of protein expression is correlated to differential cell-cell adhesion in the endomesodermal epithelium of *N. vectensis*, we assessed their role in regulating paracellular movements between ectodermal and endomesodermal epithelia by using a fluorescent tracer dye penetration assay (*Figure 2A*) (*Higashi et al., 2016*). For the purposes of these experiments, in order to avoid unwanted results related to tissue specification, cell proliferation, and cell movements, we used newly hatched juvenile polyps where the gastrodermis (endomesodermally derived) is fully differentiated.

*N. vectensis* polyps were exposed to media containing 10,000 MW fluorescent dextran (Molecular Probes, Inc.). When juvenile polyps are incubated in dextran for 5–10 min (*Figure 2B*), fluorescent dextran solution moves into the gastric cavity and then spreads into the mesoglea through the gastrodermal epithelium (*Figure 2C*) but does not enter the mesoglea through the outer ectodermally-derived epidermis (*Figure 2C and D*). These results suggest that cell-cell adhesion is differentially regulated between the epidermis and gastrodermis and the absence/disruption of AJs may compromise Septate Junctions (SJs) in the gastrodermis. Similar results were obtained in *N. vectensis* polyps when we overexpressed *Nv*Par-3::mVenus by injection of mRNA into uncleaved eggs which is normally expressed in ectodermal but not endodermal epithelial tissue (*Figure 2D and E*). However, in polyps expressing a dominant negative version of *Nv*Par-3::mVenus (dn*Nv*Par-3; microinjected into uncleaved eggs) dye penetrated between epithelial cells in both the gastrodermis and the outer epidermis (*Figures 2D, F and G*), demonstrating an ancestral role of the aPKC/Par complex in the maintenance of cell-cell adhesion and the paracellular boundary (SJs) of epithelial cells during animal development.

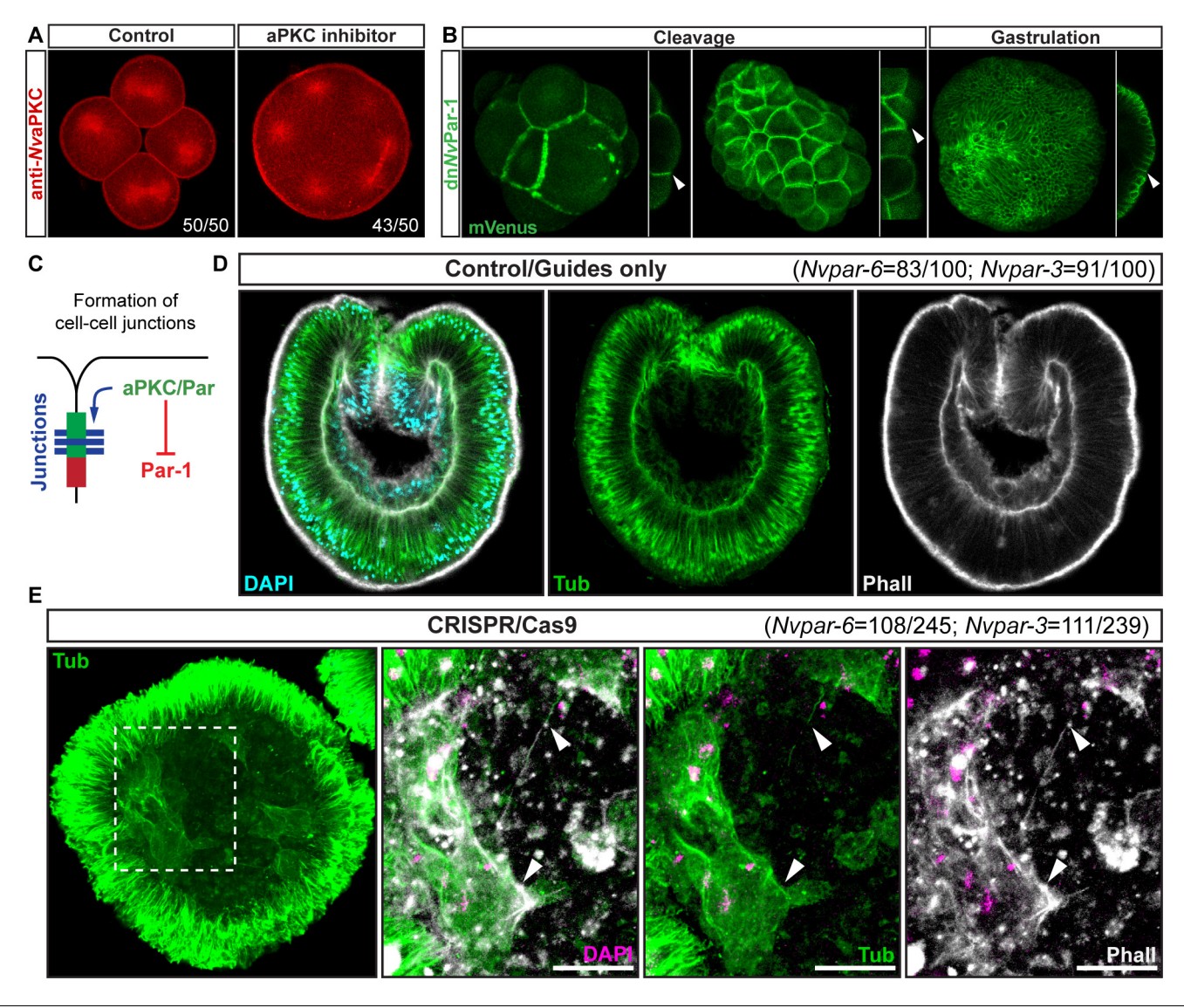

**Figure 3.** Ectodermal *Nva*PKC/Par complex polarity regulates the epithelial integrity of both ecto- and endomesoderm. (**A**) IFS for *Nva*PKC at 4 hpf showing that the aPKC inhibitor (Sigma P1614) blocks cytokinesis but not cell cycle. (**B**) *in vivo* expression of dn*Nv*Par-1 shows precocious localization to zones of cell contact during cleavage stages, well before wild-type *Nv*Par genes do. See also *Figure 3—figure supplement 1A* and *Figure 3—figure supplement 2*. (**C**) Diagram depicting the suggested the working hypothesis. (**D**) CRISPR/Cas9 knock-out for *Nv*Par-6 and *Nv*Par-3 at 48 hpf. Controls show no effect on gastrulation. Tubulin (Tub), Phalloidin (Phall), and DAPI are used as counterstains. (**E**) CRISPR/Cas9 mutants: tubulin stained low magnification of CRISPR phenotype. High magnification images from boxed region shows mesenchymal-like cells. Arrowheads indicate filopodia-like structures. Number of cases observed for each gene are shown. See also *Figure 3—figure supplement 1*, *Figure 3—figure supplements 3–6*, and *Figure 3—video 1*. Morphology is shown by DAPI, Tub, and Phall IFS. Except for 3B and 3D, all images are single optical sections from the z-stack confocal series. (**B**) and (**D**) are 3D reconstructions from a z-stack confocal series. All scale bars, 20 μm.

DOI: https://doi.org/10.7554/eLife.36740.007

The following video, source data, and figure supplements are available for figure 3:

**Figure supplement 1.** Disruption of the aPKC/Par complex in *N. vectensis* embryos.
DOI: https://doi.org/10.7554/eLife.36740.008

**Figure supplement 2.** Co-immunoprecipitation of NvPar proteins using a NvPar-1 specific antibody.
DOI: https://doi.org/10.7554/eLife.36740.009

**Figure supplement 3.** Different ectodermal thickness observed in CRISPR/Cas9 mutants.
DOI: https://doi.org/10.7554/eLife.36740.010

**Figure supplement 3—source data 1.** Measures and statistical analyses related to *Figure 3—figure supplement 3*.

*Figure 3 continued on next page*

*Figure 3 continued*

DOI: https://doi.org/10.7554/eLife.36740.011

**Figure supplement 4.** CRISPR/Cas9 mediated mutagenesis of *Nvpar-6* and *Nvpar-3*.

DOI: https://doi.org/10.7554/eLife.36740.012

**Figure supplement 4—source data 2.** Full height of the blots shown in *Figure 3—figure supplement 4*.

DOI: https://doi.org/10.7554/eLife.36740.013

**Figure supplement 5.** In vivo localization of dn*Nv*Par-6 protein at different embryonic stages.

DOI: https://doi.org/10.7554/eLife.36740.014

**Figure supplement 6.** In vivo localization of dn*Nv*Par-3 protein at different embryonic stages.

DOI: https://doi.org/10.7554/eLife.36740.015

**Figure supplement 7.** Morphometric measurements methodology.

DOI: https://doi.org/10.7554/eLife.36740.016

**Figure 3—video 1.** Related to *Figure 3E*.

DOI: https://doi.org/10.7554/eLife.36740.017

**Figure 3—animation 1.** *Figure 3E* z-stack for each separate channel.

DOI: https://doi.org/10.7554/eLife.36740.018

## The *Nv*aPKC/Par complex regulates the formation and maintenance of cell-cell junctions

Our results suggest that the absence of Par proteins in the endomesoderm is associated with changes in cell-cell adhesion complexes. Pharmacological treatment of *N. vectensis* embryos with an aPKC activity inhibitor blocks cytokinesis but not mitosis in cleaving embryos (*Figure 3A*). In addition, a dominant negative version of *Nv*Par-1 (dn*Nv*Par-1), that lacks its kinase domain, localizes only to the cortex of cell‑cell contacts (*Figure 3B*). Since *Nv*Par-1 is phosphorylated by *Nv*aPKC (*Figure 3—figure supplement 2*), we predict that, as in other systems, dn*Nv*Par-1 could be phosphorylated by aPKC but would not phosphorylate the aPKC/Par complex (*Vaccari et al., 2005*; *Böhm et al., 1997*). Thus, dn*Nv*Par-1 can localize to the cell cortex where aPKC may be inactive. These results together suggest that the formation of cell‑cell contacts is regulated by the activity of the aPKC/Par complex in *N. vectensis* embryos (*Figure 3C*).

We further tested this hypothesis by using genome editing by CRISPR/Cas9 targeting *Nvpar-6* and *Nvpar-3* genes (*Figure 3D*). We did not observe any effects on the embryo until 36 hpf at 16°C (late blastula stage), indicating the activity of maternally loaded proteins up until that stage. When *Nv*Par-6 and *Nv*Par-3 are mutated, the ectodermal epithelium loses its integrity, presenting changes in thickness (*Figure 3—figure supplement 1B* and *Figure 3—figure supplement 3*), and interestingly, the endomesoderm (which does not express these proteins) generates cells with mesenchymal-like morphotypes that are never normally seen in this species (*Figure 1D and E*). In *Nvpar-6* and *Nvpar-3* mutant embryos, we also observed the disruption of microtubules and actin cytoskeleton (*Figure 3—figure supplement 4B*), and AJs (visualized with the ß-catenin antibody in *Figure 4*) that confirms our previous observations of their role in regulating ectodermal cell polarity. Although it was difficult to dissect significant changes in the expression of germ layer markers (e.g. *Nvbra*, *Nvsnail*, *NvSix3/6*, and *Nvfz10*) from the morphological changes associated with epithelial integrity when these genes were disrupted (*Figure 3—figure supplement 4E*), it is clear that the primary defect in NvPar3 KO were aspects of cell adhesion and not cell type specification. Similar results were obtained when we overexpressed the mRNA encoding for a dominant negative version *Nv*Par-6 (dn*Nv*Par-6) and *Nv*Par-3 (dn*Nv*Par-3) into *N. vectensis* eggs (*Figure 3—figure supplement 1* and *Figure 3—figure supplements 5* and *6*). However, dominant negative effects on the injected embryos were observed at earlier stages (10–12 hpf) than the CRISPR/Cas9 mutants (zygotic expression) because the mutant proteins compete with the wild type proteins (maternally loaded). Hence, in these experiments, embryonic lethality (90%) and cell death were higher.

## The *Nv*aPKC/Par complex regulates transepithelial signaling

One surprising observation from the experiments described above show that the changes observed in the ectodermal and endodermal epithelium after disrupting *Nv*Par-6 and *Nv*Par-3 (*Figure 3*) suggests some sort of trans-epithelial regulation of cell‑cell adhesion (most likely involving AJs) because these Par genes are not expressed in the endomesoderm. The polarizing activity of the aPKC/Par

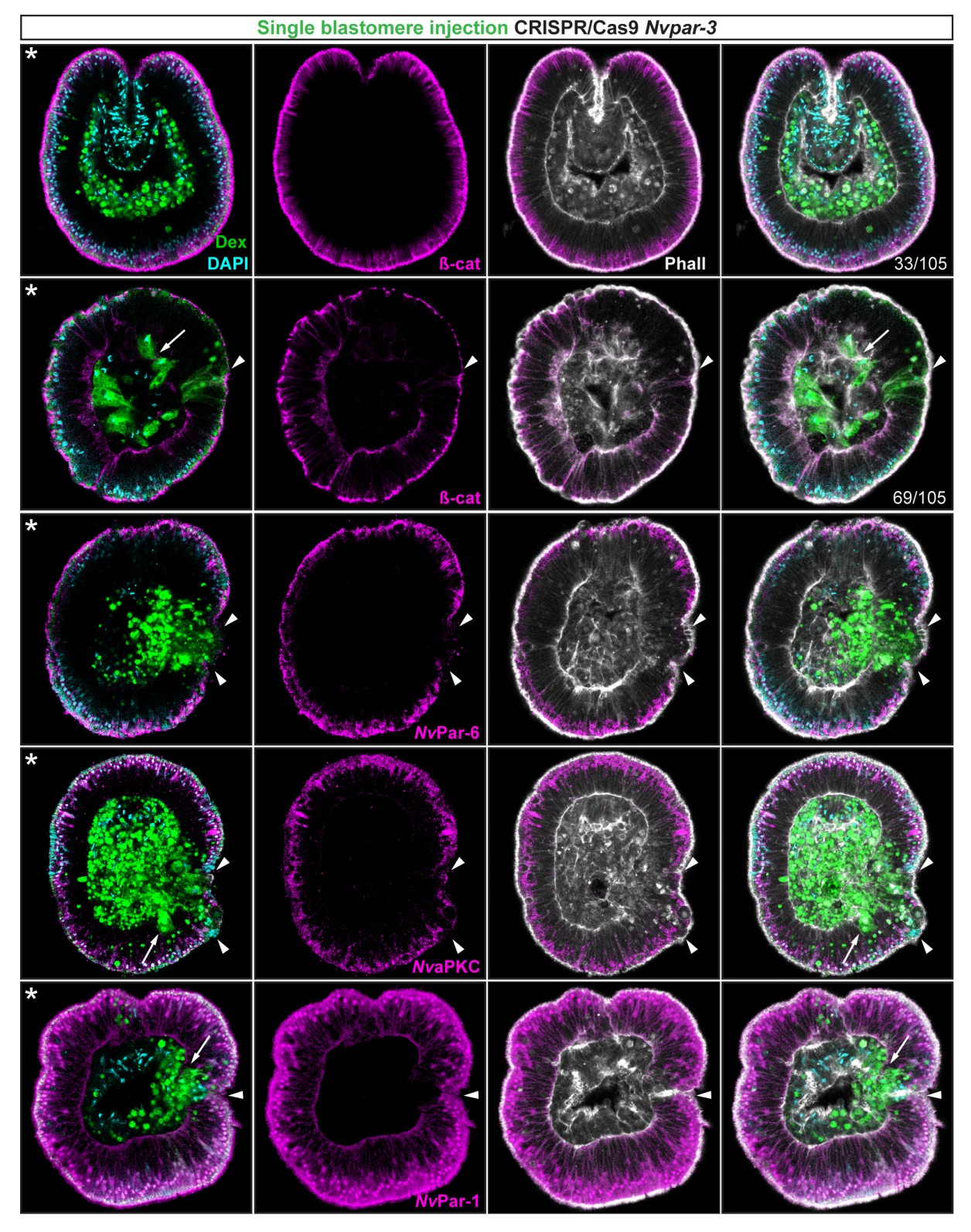

**Figure 4.** Ectodermal- but not endomesodermal-lineages are affected by single injected-blastomere CRISPR/Cas9 *Nvpar-3* knock-outs. IFS for ß-catenin (ß-cat), *Nv*Par-6, *Nva*PKC, and *Nv*Par-1 in single injected-blastomere CRISPR/Cas9 *Nvpar-3* knock-outs at 40 hpf. Streptavidin-Biotin TxRed Dextran (Dex) is shown in green. Arrowheads indicate the absence of the protein and disrupted epithelium. Arrows indicate bottle-like shape cells. *
*Figure 4 continued on next page*

*Figure 4 continued*

indicate the orientation of the site of gastrulation. See also *Figure 4—figure supplements 1* and *2*, and *Figure 4—video 1*. Morphology is shown by DAPI and Phalloidin. All images are single optical sections from the z-stack confocal series.

DOI: https://doi.org/10.7554/eLife.36740.019

The following video and figure supplements are available for figure 4:

**Figure supplement 1.** Diagram depicting the hypothesis addressed in *Figure 4*.

DOI: https://doi.org/10.7554/eLife.36740.020

**Figure supplement 2.** Immunofluorescent staining for ß-catenin (ß-cat) in single injected-blastomere control and CRISPR/Cas9 *Nvpar-3* knock-outs at 40 hpf.

DOI: https://doi.org/10.7554/eLife.36740.021

**Figure 4—video 1.** z-stack series of CRISPR phenotype (ß-catenin panel).

DOI: https://doi.org/10.7554/eLife.36740.022

**Figure 4—animation 1.** *Figure 4* z-stack for ß-catenin panel.

DOI: https://doi.org/10.7554/eLife.36740.023

**Figure 4—animation 2.** *Figure 4* z-stack for NvPar-6 panel.

DOI: https://doi.org/10.7554/eLife.36740.024

complex in the ectoderm is thus necessary to maintain the integrity of both ecto- and endodermal epithelia during cellular movements associated with gastrulation.

To assess whether the observed phenotypes on cell-cell adhesion are related to non-autonomous cell regulation (trans-epithelial interactions), we repeated the above experiments randomly injecting single blastomeres at 3–4 hpf (8–16 cell-stage) to make mutant clones in an otherwise wild type background (*Figure 4*). In these experiments, only the cell-lineage of the injected blastomere would be affected and would exhibit defective cell-cell adhesion in an otherwise undisturbed wild-type background. If endomesodermal cells derived from an injected blastomere display fibroblast/mesen-chymal cell morphology, it would indicate that the organization of the endomesodermal epithelium is not dependent on the ectoderm but, rather, an intrinsic cell-autonomous activity of the aPKC/Par complex (*Figure 4—figure supplement 1*). Our results show that only ectodermal- but not endome-sodermal-lineages are affected by these mutations (*Figure 4* and *Figure 4—figure supplement 2*). Presumptive ectodermal cells derived from an injected blastomere fail to maintain AJs (and poten-tially SJs) and the resulting clone of epithelial cells loses its structural integrity inducing cell extru-sion. In contrast, presumptive endomesodermal cells derived from an injected blastomere develop into a normal endomesodermal epithelium (*Figure 3F*). Our results complement the work of (*Kirillova et al., 2018*) and demonstrate that the proper cell-cell adhesion of the ectodermal layer somehow regulates trans-epithelially the integrity of the endomesodermal layer. This regulation may maintain the tension between cells during invagination at gastrula stages, or, in conjunction with the extracellular matrix (ECM) and basal cues, it may influence signaling patterns necessary to organize epithelial layers during *N. vectensis* embryogenesis.

## Interaction between the *Nv*aPKC/Par complex and the canonical Wnt signaling pathway

### *Nv*aPKC/Par complex regulates ß-catenin localization

In bilaterians, AJs recruit members of the aPKC/Par complex and the direct interaction between Par-3 and aPKC/Par-6 is required for the maintenance and maturation of AJs (*Ohno et al., 2015*; *Ragkousi et al., 2017*). AJs are characterized by the binding between cadherins and ß-catenin: cad-herins sequester ß-catenin from the cytoplasm to the cortex, making it unavailable for nuclear signal-ing and endomesoderm specification (*Wikramanayake et al., 2003*; *Kumburegama et al., 2011*)[24,41]. Therefore, using ß-catenin as a marker for AJs, we separately co-injected *Nv*Par-3::mVe-nus or a mutated dn*Nv*Par-3::mVenus, with *Nv*ß-catenin::RFP into uncleaved zygotes. We observed cortical co-localization of *Nv*Par-3 and *Nv*ß-catenin at the cell boundaries in the ectodermal epithe-lium of embryos co-injected with *Nv*Par-3::mVenus and *Nv*ß-catenin::RFP (*Figure 5A*). However, in embryos co-injected with *Nv*ß-catenin::RFP and dn*Nv*Par-3::mVenus, we observed an alteration of the sub-cellular expression of *Nv*ß-catenin::RFP in all cells due to the translocation of ß-catenin from the cortical AJs into cell nuclei (*Figure 5A*).

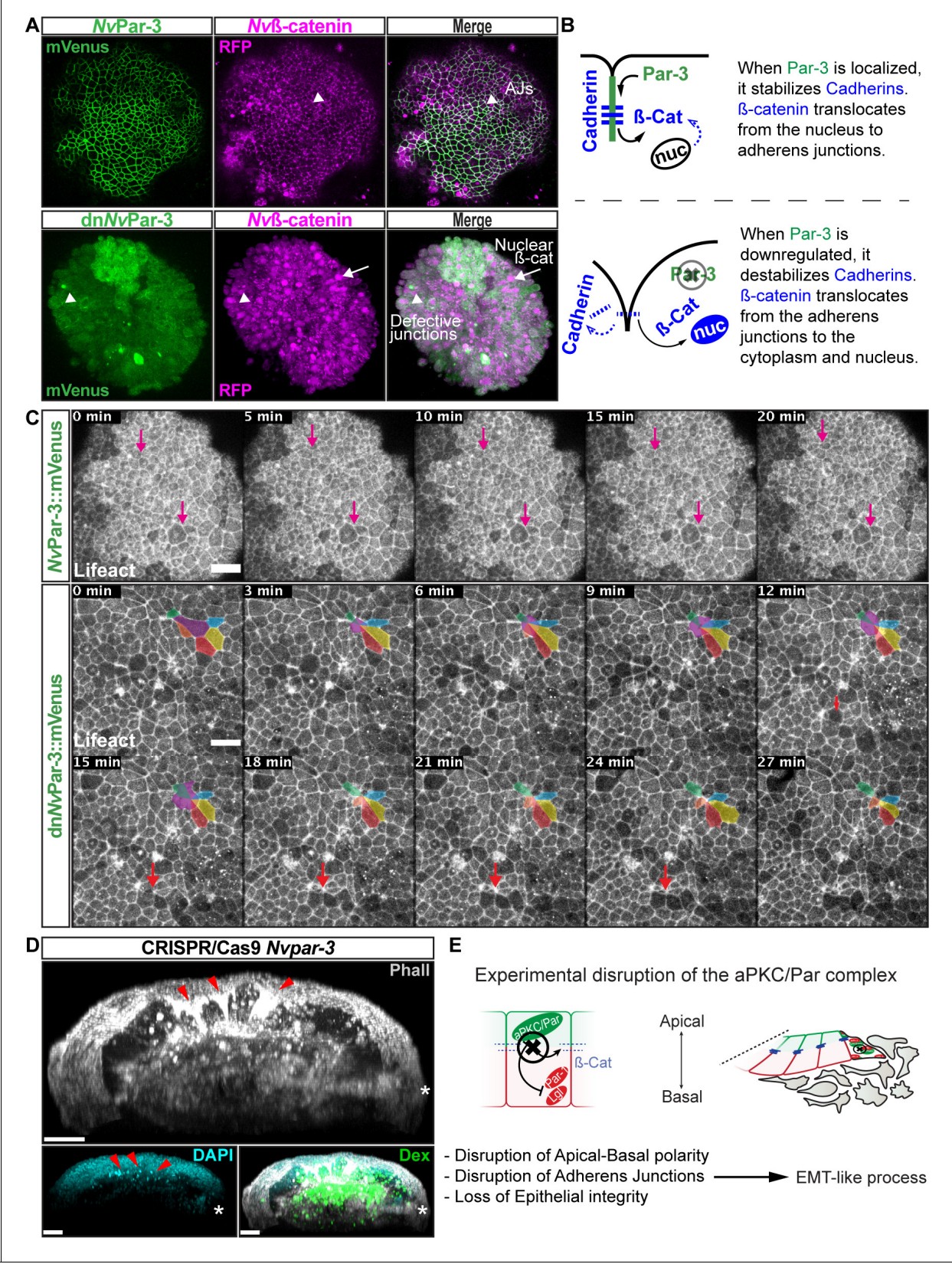

**Figure 5.** NvaPKC/Par complex regulates ß-catenin localization and cell attachment. (**A**) In vivo co-localization of NvPar-3venus co-injected with Nvß-cateninRFP, and dnNvPar-3venus co-injected with Nvß-cateninRFP. Arrowheads indicate junctions (AJs). Arrows indicate nuclear ß-catenin. (**B**) Diagram

Figure 5 continued on next page

*Figure 5 continued*

of the suggested interpretation for A. (**C**) In vivo time series of ectodermal epithelial layers of embryos injected with *Nv*Par-3venus and dn*Nv*Par-3venus mRNA demonstrating epithelial delamination in the absence of functional *Nv*Par3. Lifeact::mTq2 mRNA was co-injected to visualize cell boundaries. Pink arrows indicate the absence cell detachments. A subset of cells was artificially colored. The purple cell detaches from the epithelium and the red arrow indicates a second celldetachment. See also *Figure 5—video 1*. (**D**) IFS of an embryo in which a single blastomere was injected with *Nv*Par-3 guide RNAs and Cas9 and green dextran. Red arrowheads indicate the apical constriction and delamination of ectodermal cells in the mutated clone of cells. Note the different layers of nuclei stained with DAPI. Asterisks indicate the site of gastrulation. (**E**) Diagram of the suggested interpretation for D. All images are 3D reconstructions from a z-stack confocal series. All scale bars, 20 μm.

DOI: https://doi.org/10.7554/eLife.36740.025

The following video and figure supplement are available for figure 5:

**Figure supplement 1.** Apical junctions are regulated by GSK-3ß in epithelial cells during gastrulation in *N. vectensis* embryos.
DOI: https://doi.org/10.7554/eLife.36740.026
**Figure 5—video 1.** Related to *Figure 5C*.
DOI: https://doi.org/10.7554/eLife.36740.027

In addition, results from *N. vectensis* embryos treated with 5 μm 1-azakenpaullone (AZ; an inhibitor of GSK-3ß and a canonical Wnt agonist) suggest that GSK-3ß stabilizes AJs of epithelial cells in *N. vectensis* embryos (*Figure 5—figure supplement 1*). We observed an expansion of the expression domain of Par-6 (*Figure 5—figure supplement 1*) and a stabilization of AJs (labeled with ß-catenin in *Figure 5—figure supplement 1*) in endomesodermal cells of treated embryos, which was never observed in control embryos.

Interestingly, the association between the nuclearization of ß-catenin (canonical Wnt signaling pathway) and the Par system has been poorly studied. Two studies, one in *Drosophila* (*Sun et al., 2001*) and the another in *Xenopus* (*Ossipova et al., 2005*) embryos, have shown by immunoblotting that the kinase Par-1 is associated with Dishevelled protein and might act as a positive regulator of Wnt signaling. Here, we show *in vivo* embryonic evidence suggesting that *Nv*Par-3 (whose cortical localization is normally inhibited by Par-1) recruits *Nv*ß-catenin protein and stabilizes its localization at the apico-lateral cortex of ectodermal cells through the formation of AJs. Furthermore, the putative disassembly of the aPKC/Par complex induced by dn*Nv*Par-3 overexpression, induces the nuclearization of *Nv*ß-catenin protein (*Figure 5A*) due to its cytosolic availability caused by AJs disruption. Strikingly, we also observed the extrusion of individual cells from the ectodermal epithelium of dn*Nv*Par-3 treated-embryos (*Figure 5C*) and single injected-blastomere CRISPR/Cas9 *Nv*Par-3 knock-out (*Figure 5D*). This suggests that these treatments induce EMT-like processes, not observed under control conditions (*Figure 5C*).

Thus, our data suggest that preexisting mechanisms downstream to the induction of EMT may have been redeployed to segregate layers during the evolution to bilaterians. Bringing the question whether or not endomesodermal genes would induce similar effects when they are expressed in *N. vectensis* embryos.

We have recently showed that *Nvbrachyury* regulates apicobasal polarity of epithelial cells in *N. vectensis* embryos (*Servetnick et al., 2017*). We, therefore, examined the role of *Nvsnail* genes on the localization of ß-catenin, components of the Par system, and the stabilization of AJs. Our hypothesis is that expression of *N. vectensis* snail genes would destabilize AJs and induce the nuclearization of ß-catenin in ectodermal epithelial cells.

## *Nvsnail* genes induce the translocation of *Nv*ß-catenin from AJs to the cytoplasm

*N. vectensis* has two *snail* genes, *Nvsnail-A* and *Nvsnail-B*, which are both expressed in the endomesodermal plate prior to and throughout the gastrulation process, and which define the boundary between gastrodermis and ectodermal pharynx (*Magie et al., 2007*; *Röttinger et al., 2012*; *Amiel et al., 2017*). To determine the role of *Nvsnail* genes on ß-catenin nuclearization, we co-injected the mRNA of *Nv*Snail-A::mCherry, *Nv*Snail-B::mCherry, and *Nv*ß-catenin::GFP into uncleaved eggs. The overexpression of both proteins *Nv*Snail-A::mCherry and *Nv*Snail-B::mCherry together induce the ectopic translocation of *Nv*ß-catenin::GFP to the nuclei of ectodermal cells (*Figure 6A*). This treatment also delocalizes *Nv*Par-3 from the cell cortex when both *Nv*Snail::mCherry proteins are co-expressed with *Nv*Par-3::mVenus (*Figure 6B*).

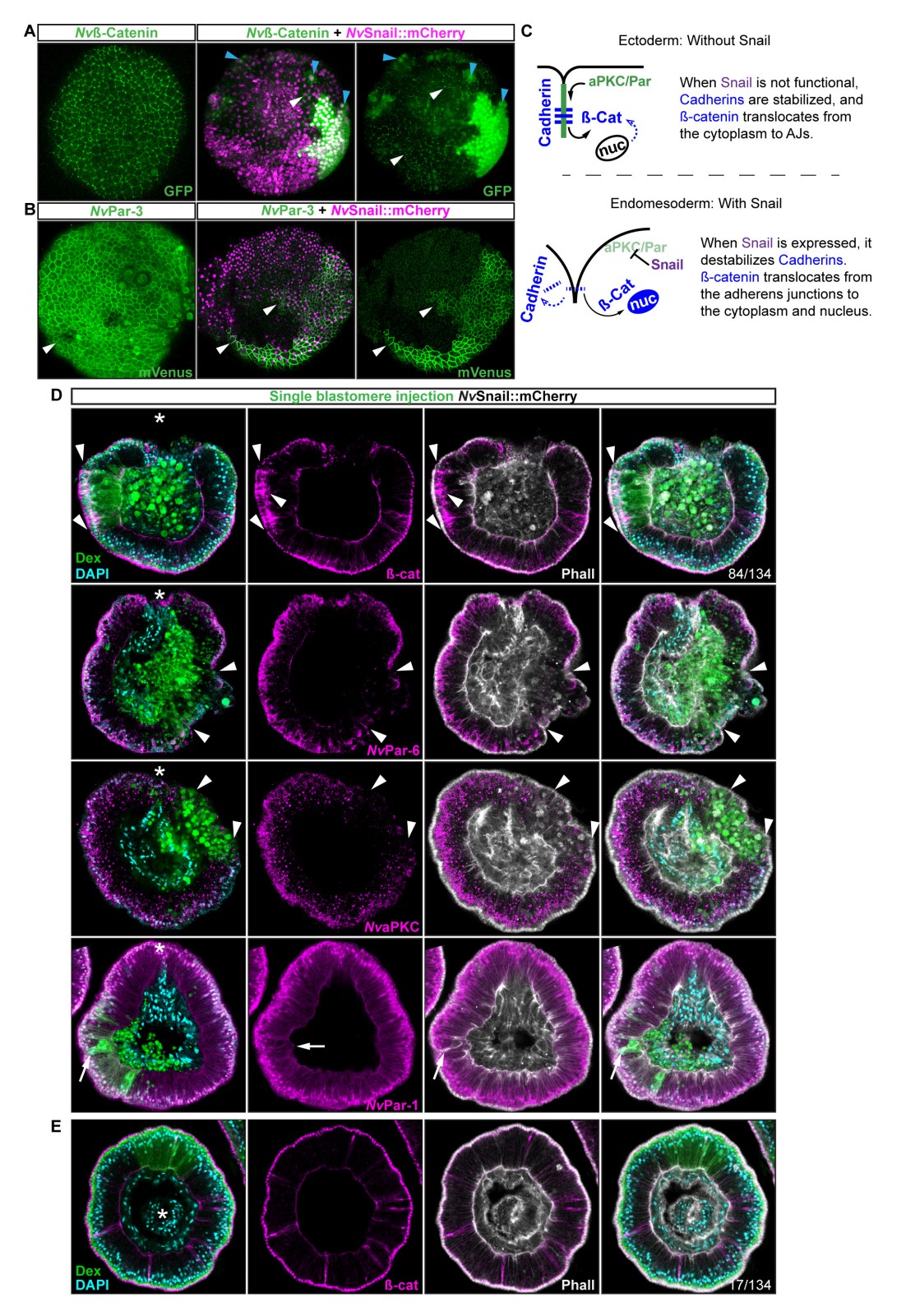

**Figure 6.** *Nvsnail* genes induce the translocation of *Nv*ß-catenin and the disruption of epithelial integrity. (**A**) *in vivo* localization of *Nv*ß-cateninGFP co-injected with both *Nv*Snail-A::mCherry and *Nv*Snail-B::mCherry mRNA together in zygotes at 40 hpf. White arrowheads indicate AJs. Patched patterns of cytosolic and nuclear ß-catenin (Blue arrowheads) were observed. (**B**) *in vivo* localization of *Nv*Par-3::mVenus co-injected with both *Nv*Snail-A:: mCherry and *Nv*Snail-B::mCherry mRNA together at 40 hpf. Patched patterns of AJs (White arrowheads) were observed. (**C**) Diagram depicts the

*Figure 6 continued on next page*

*Figure 6 continued*

suggested interpretation for A and B. (**D**) IFS for ß-catenin (ß-cat), *Nv*Par-6, *Nv*aPKC, and *Nv*Par-1 in embryos at 40 hpf where *Nv*Snail-A::mCherry and *Nv*Snail-B::mCherry mRNA were overexpressed together into a single ectodermal blastomere lineage (followed by green Streptavidin-TxRed Dextran (Dex). Arrowheads indicate the absence of the protein, cytosolic ß-cat, and disrupted epithelium. Arrows indicate bottle-like shape cells. (**E**) IFS for ß-cat in embryos at 40 hpf where *Nv*Snail-A::mCherry and *Nv*Snail-B::mCherry mRNA were overexpressed together into a single blastomere lineage and no affects were observed. See also *Figure 6—figure supplement 1*, *Figure 6—video 1*, and *Figure 6—video 2*. *site of gastrulation. Morphology is shown by DAPI and Phall IFS. Except from 6A and 6B, all images are single optical sections from the z-stack confocal series. (**A**) and (**B**) are 3D reconstructions from a z-stack confocal series.

DOI: https://doi.org/10.7554/eLife.36740.028

The following video and figure supplement are available for figure 6:

**Figure supplement 1.** *Nvsnail-A and Nvsnail-B* together regulate epithelial integrity.

DOI: https://doi.org/10.7554/eLife.36740.029

**Figure 6—video 1.** Related to *Figure 6E*.

DOI: https://doi.org/10.7554/eLife.36740.030

**Figure 6—video 2.** Related to *Figure 6E*.

DOI: https://doi.org/10.7554/eLife.36740.031

**Figure 6—animation 1.** *Figure 6D* z-stack for NvPar-6 panel.

DOI: https://doi.org/10.7554/eLife.36740.032

**Figure 6—animation 2.** *Figure 6D* z-stack for NvPar-1 panel.

DOI: https://doi.org/10.7554/eLife.36740.033

To determine the role of *Nvsnail* genes on cell adhesion/epithelial polarity, we randomly injected single blastomeres at the 8–32 cell-stage with mRNA from both *Nv*Snail-A::mCherry and *Nv*Snail-B::mCherry together. The fluorescent dextran that was co-injected with the mRNAs could be used to detect the clones where the over-expression of the co-injected mRNAs occurred in a 'wild-type' background (*Figure 6D*). Similar to the *Nvpar-3* knock-out (*Figure 4*), the expression of *Nvsnail* genes is sufficient to induce the degradation of Par proteins and AJs (ß-catenin) from the ectoderm and disrupts its epithelial integrity; however, nuclear ß-catenin was not observed under these treatments (*Figure 6D*). Thus, nuclear *Nv*ß-catenin::GFP observed *in vivo* when we overexpressed *Nv*Snail proteins (*Figure 6A*) is a consequence of the high cytosolic availability generated by its ectopic over-expression and release from AJs.

Interestingly, not every ectodermal cell was affected by these treatments even though all of the cells expressed the injected mRNAs (*Figure 6A and E*, and *Figure 6—figure supplement 1*). This patched pattern suggests that the response to *Nvsnail* over-expression is spatially regulated. These results suggest that the role of *Nvsnail* genes on AJs and apicobasal cell polarity is constrained to the site of gastrulation in *N. vectensis* embryos under natural conditions, and that these genes may be required for gastrulation movements. Therefore, we predicted that ß-catenin (AJs) and Par proteins will be retained in the cells of the *N. vectensis* endomesodermal plate if both *Nvsnail* genes are disrupted.

### *Nvsnail* genes downregulate apicobasal cell polarity and AJs in the endomesodermal epithelium of *N. vectensis* embryos

The *snail* genes temporally down-regulate E-cadherin during mesoderm segregation and EMT in bilaterian animals (*Lim and Thiery, 2012*). As we have shown here, as well as in previous studies (*Magie et al., 2007*; *Magie and Martindale, 2008*), the cells comprising the endomesodermal plate lose their cell-cell adhesion during gastrulation in *N. vectensis* embryos. It may be possible that temporal regulation of endomesodermal patterning might act upon the AJs. Our data suggest that once gastrulation is complete and the pharynx forms, components of the Par system and the ß-catenin components of the AJs are degraded from both the cortex and cytoplasm of endomesodermal cells (*Figure 1* and *Figure 1—figure supplement 1*). Hence, it could be possible that *Nvbrachyury* induces the disruption of apicobasal polarity (*Servetnick et al., 2017*), remnant AJs maintain the endomesodermal-plate cells together, and *Nvsnail* genes degrades and prevents the reassembly of AJs in the endomesoderm of *N. vectensis*.

To address these issues, we used CRISPR/Cas9 knock-out of *Nvsnail-A* and *Nvsnail-B* genes together to inhibit zygotic function of these genes and investigate their role on the temporal

regulation of AJs and cell polarity. In CRISPR/Cas9 mutants, the endomesodermal plate forms but it does not migrate further than its first invagination during gastrulation (*Figure 7*). Furthermore, AJs (labeled with ß-catenin) and apical Par proteins (labeled with anti*Nv*Par-6 and anti*Nva*PKC) are retained at the apical cortex of the cells of the endomesodermal plate (*Figure 7B and C* and *Figure 7—figure supplement 1*). Surprisingly, *Nv*Par-1 and *Nv*Lgl were not detected in those cells (*Figure 7C*), suggesting that the degradation of these basolateral proteins precede or do not depend on the activity of the *Nvsnail* genes. This suggests that *Nvsnail* regulates apical cell-polarity, AJs turnover, and the migration ('zippering') but not the invagination of the endomesodermal plate during gastrulation of *N. vectensis* embryos (*Figure 7D*). Interestingly, the invagination of the endomesodermal plate (controlled by the Wnt/PCP pathway) is uncoupled from its specification in *N. vectensis* embryos (*Kumburegama et al., 2011*; *Wijesena et al., 2011*), which is consistent with our observations.

## Discussion

### AJs are down-regulated in mesoderm and neural crest of bilaterian animals

The segregation of different germ layers during embryogenesis of many bilaterian animals is carried out by similar cellular mechanisms. EMT is a shared mechanism utilized by mesoderm, neural crest cell (NCC), and tumorigenesis to delaminate cells in bilaterian animals (triploblastic animals). During EMT, the nuclearization of ß-catenin induces the expression of 'endomesodermal' genes like *brachyury* and *snail* (*Acloque et al., 2009*). The expression of these genes downregulates epithelial cadherins, disrupts apicobasal polarity (mediated by the aPKC/Par complex), disassembles AJs, and induces changes in cytoskeleton organization (*Acloque et al., 2009*; *Lim and Thiery, 2012*). A rearrangement of the actin-myosin cytoskeleton induces apical constriction of cells (generating a bottle-like shape), which detach from the epithelial sheet, break down the basal membrane, and invade a specific tissue as mesenchymal cells (*Acloque et al., 2009*; *Lim and Thiery, 2012*; *Ohsawa et al., 2018*).

Interestingly, mesoderm formation, tumorigenesis, and EMT have never been described as natural processes during *N. vectensis* (a diploblastic animal) embryogenesis. During *N. vectensis* gastrulation (*Magie et al., 2007*; *Tamulonis et al., 2011*), cells around the edge of the blastopore at the animal pole (which expresses *Nvbrachyury*) acquire a bottle-like shape by apical constriction, leading to epithelial buckling and the invagination of presumptive endomesoderm (which expresses *Nvsnail*). However, throughout this process the endomesoderm remains as a monolayer of epithelial cells and individual mesenchymal cells never detach and invade the blastocoel.

We have shown that by disrupting the aPKC/Par complex (apicobasal cell-polarity) in *N. vectensis* (*Figures 3*, *4* and *5*), we are able to convert cells from the endomesodermal epithelium into mesenchymal-like cells, translocate *Nv*ß-catenin (*Figure 5A*), and emulate EMT-like processes (apical constriction and individual cell-detachments) in the ectodermal epithelium of *N. vectensis* treated-embryos (*Figure 5C and D*). These results demonstrate that the cnidarian *N. vectensis* possesses mechanisms necessary to segregate individual germ layers (e.g. mesoderm and NCC) described in bilaterians; however, they do not do it.

Our working hypothesis is that the *N. vectensis* embryo is composed of two independent morphogenetic modules that are integrated and organized by the pharynx (*Figure 7D*). The first observation is that the ectoderm, whose apicobasal polarity (and thus, AJs and epithelial integrity) is regulated by *Nvbrachyury* that promotes ectodermal epithelial morphogenesis and pharynx formation (*Servetnick et al., 2017*), and the second module is generated by endomesodermal differentiation and cell-movements that are regulated by *Nvsnail* genes. This is supported by the expression *Nvbrachyury* in *Nvsnail* knock-out embryos (*Figure 7—figure supplement 2*), and *Nvsnail* knock-out phenotypes where ectodermal pharynx develops normally but no clear endomesoderm is formed (*Figure 7—figure supplement 1*). Additional work is required to elucidate any differences in function between *Nvsnail-A* and *Nvsnail-B* genes, however, both modules are specified by nuclear ß-catenin (*Röttinger et al., 2012*), suggesting that the nuclear ß-catenin (maternal) shift from the animal pole in cnidarians to the vegetal pole in bilaterians is mechanistically plausible and sufficient to re-

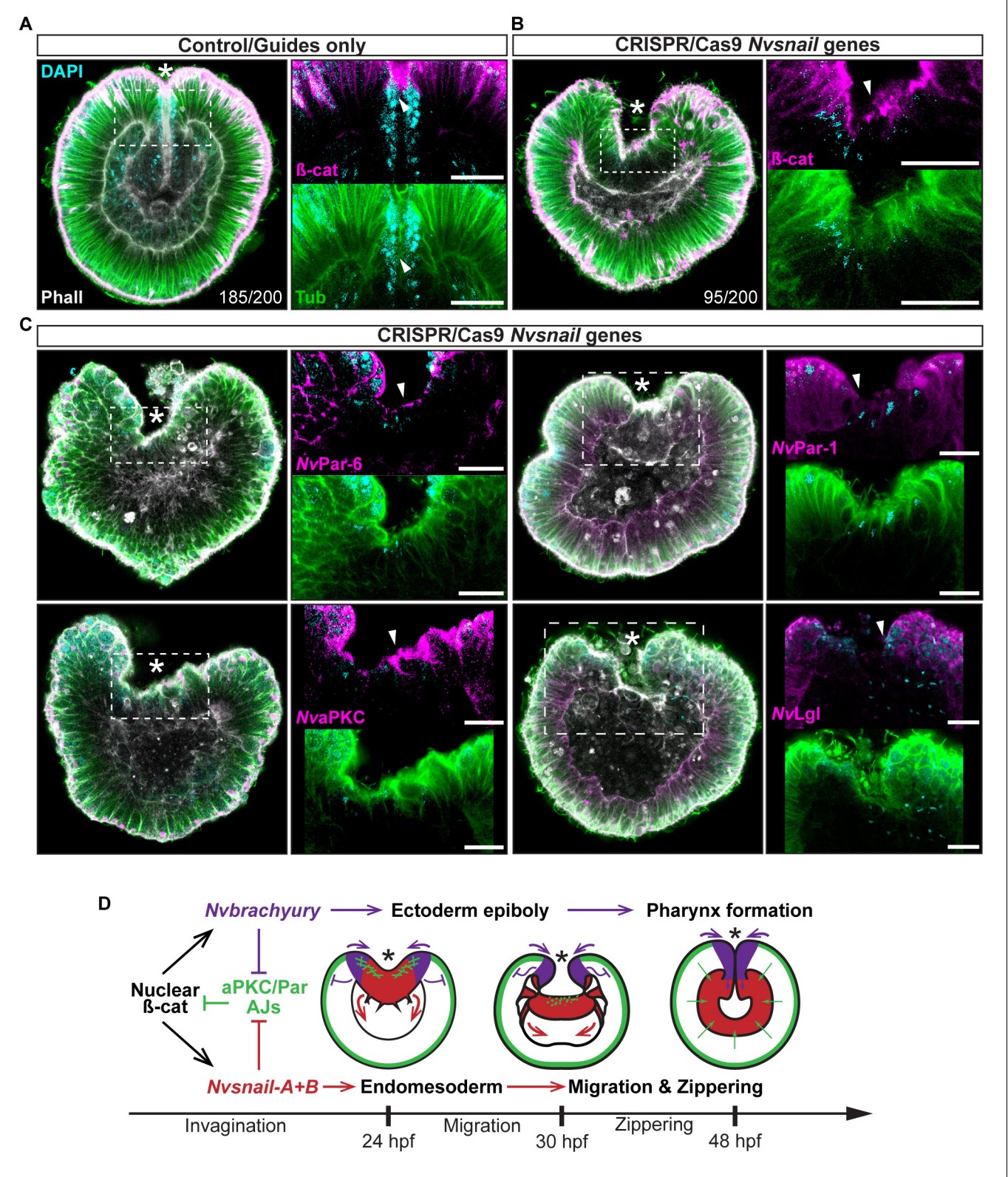

**Figure 7.** *Nvsnail* genes prevents the reassembly AJs and *Nv*aPKC/Par polarity allowing endomesodermal migration. (A) Embryo wide CRISPR/Cas9 knock-out guides controls show no effect neither on gastrulation nor AJs (ß-catenin) localization. (B) Embryo wide CRISPR/Cas9 knock-out for both *Nvsnail-A* and *Nvsnail-B* at 40 hpf showing that AJs are retained in presumptive endomesodermal region similar to ectodermal cells. High magnification images from boxed region (endomesodermal plate) are shown on the right. (C) Embryo wide CRISPR/Cas9 knock-out for both *Nvsnail-A* and *Nvsnail-B*

*Figure 7 continued on next page*

*Figure 7 continued*

at 40 hpf showing that *NvPar-6* and *NvaPKC* proteins are retained in in presumptive endomesodermal region similar to ectodermal cells. *NvPar-1* and *NvLgl* were not detected in endomesodermal cells. High magnification images from boxed region (endomesodermal plate) are shown on the right. (**D**) Graphical summary of the observed results with previous published data (**Servetnick et al., 2017**). See also *Figure 7—figure supplements 1* and *2*. Morphology is shown by DAPI, Tubulin, and Phalloidin. All images are single optical sections from the z-stack confocal series. Arrowheads indicate protein localization. *site of gastrulation. All scale bars, 20 μm.

DOI: https://doi.org/10.7554/eLife.36740.034

The following source data and figure supplements are available for figure 7:

**Figure supplement 1.** *Nvsnail-A and Nvsnail-B* together regulate AJs.

DOI: https://doi.org/10.7554/eLife.36740.035

**Figure supplement 2.** CRISPR/Cas9 mediated mutagenesis of *Nvsnail-A and Nvsnail-B*.

DOI: https://doi.org/10.7554/eLife.36740.036

**Figure supplement 2—source data 1.** Full height of the blots shown in *Figure 7—figure supplement 2*.

DOI: https://doi.org/10.7554/eLife.36740.037

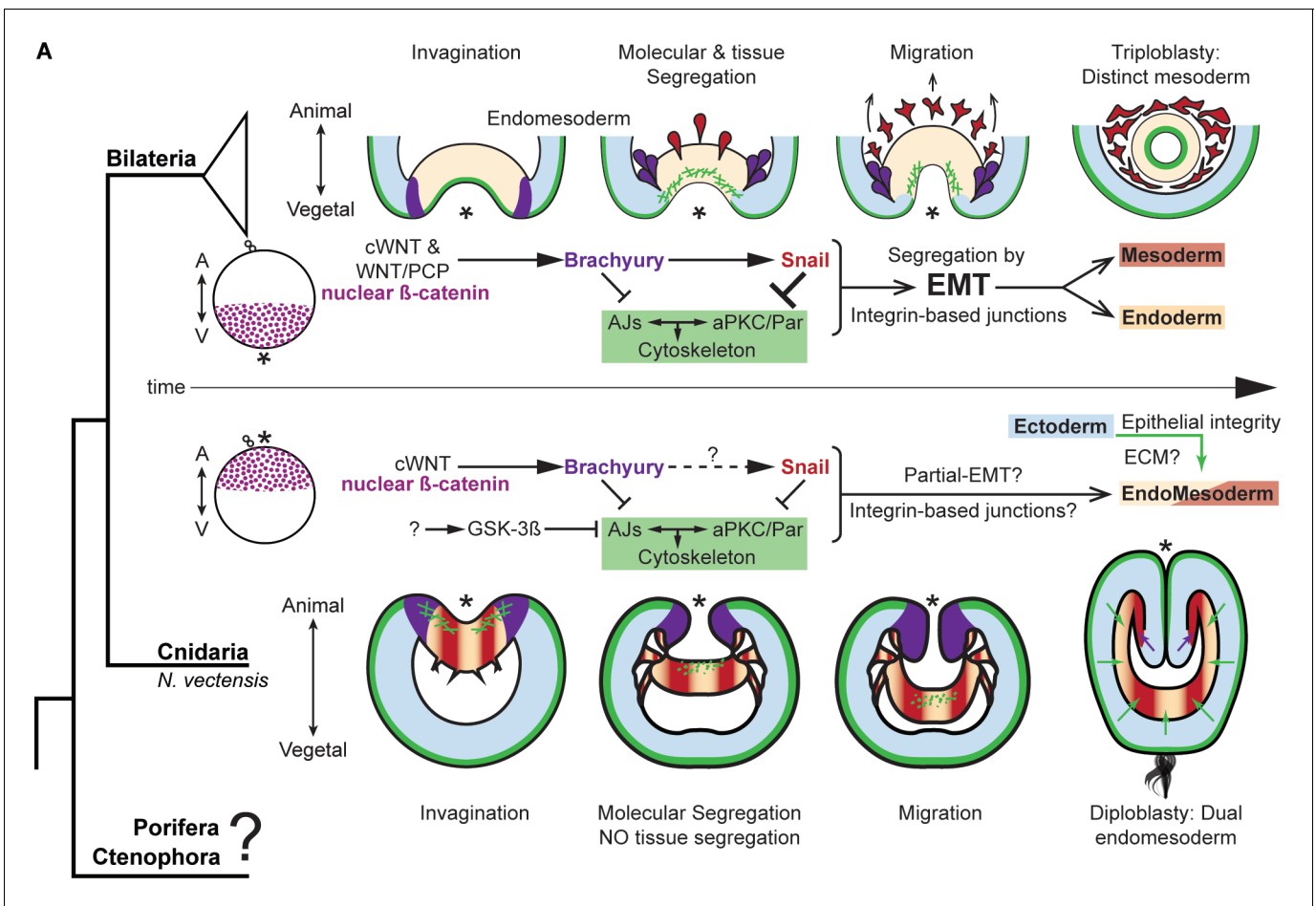

**Figure 8.** The differences between epithelial structure in ectoderm and endomesoderm in *N. vectensis* embryos are due to the lack of mechanisms to segregate a distinct mesoderm. Diagram depicting key cellular and molecular mechanisms involved during gastrulation of bilaterian and *N. vectensis* (a cnidarian) embryos. See also *Figure 8—figure supplement 1*.

DOI: https://doi.org/10.7554/eLife.36740.038

The following figure supplement is available for figure 8:

**Figure supplement 1.** Suggested model for mesoderm specification in Metazoa.

DOI: https://doi.org/10.7554/eLife.36740.039

specify the site of gastrulation and germ-layers along the animal-vegetal axis during Metazoan evolution (*Martindale and Lee, 2013*; *Lee et al., 2007*).

## The dual identity and collective migration of the endomesodermal cells

Bilaterian-EMT has been a focus of study for decades as a mechanism to segregate different cell layers involved in a variety of different normal and pathological biological processes (*Ohsawa et al., 2018*; *Nieto et al., 2016*). This process appears to depend on the fine regulation of *snail* expression levels and their temporal activity. For example, during NCC migration, cells display 'partial-EMT' where cells remain attached to several neighboring cells but their apicobasal polarity and AJs are down-regulated, allowing collective-cell migration (*Weng and Wieschaus, 2017*; *Nieto et al., 2016*; *Lee et al., 2006*; *Theveneau and Mayor, 2013*; *Ribeiro and Paredes, 2014*). Our data suggest that 'partial-EMT' may be the mechanism by which the endomesodermal epithelium migrates into the blastocoel in *N. vectensis* embryos during normal gastrulation (*Figure 8*). In this scenario, upstream factors that regulate *snail* transcription may be critical for this process.

In bilaterian animals, there are many other pathways in addition to the canonical Wnt pathway that activate *snail* transcription and induce the disruption of AJs and apicobasal cell polarity. For example, TGFß, BMP, NANOS, FGF, and MEK/ERK/ERG take on roles during the specification of mesoderm, NCC migration, tumorigenesis, and other EMT-related processes (*Lim and Thiery, 2012*; *Nieto et al., 2016*; *Barrallo-Gimeno and Nieto, 2005*). Concordantly in *N. vectensis* embryos, cells of the pharyngeal and endomesodermal tissues express components of all these pathways (*Röttinger et al., 2012*; *Amiel et al., 2017*; *Extavour et al., 2005*; *Matus et al., 2006*, *2007*; *Wijesena et al., 2017*) that may modify their cellular characteristics. For example, one cadherin (NvCDH2 *Clarke et al., 2016*: 1g244010), and kinases that modify tubulin and histones are differentially regulated between ecto- and endomesodermal epithelium (*Wijesena et al., 2017*).

In conclusion, *N. vectensis* has both up and downstream cellular and molecular mechanisms associated with EMT described in bilaterians. However, *N. vectensis* does not segregate a distinct mesodermal germ layer nor display EMT under natural conditions. In bilaterians, this mechanism must have evolved to segregate mesodermal cells from the endoderm to retain the tight cell-cell junctions required in endodermal epithelia. Interestingly, mesoderm segregation via EMT in *Drosophila* takes place after epithelial folding in response to *snail* expression. In these embryos, contractile myosin enhances the localization of AJs and Par-3 in the presumptive mesoderm and prevents their downregulation by Snail, thus delaying EMT (*Weng and Wieschaus, 2016*, *2017*). Furthermore, the overexpression of Snail in *Drosophila* embryos is sufficient to disassemble ectodermal-AJs, but mesodermal-AJs are maintained by actomyosin contraction that antagonize Snail effects (*Weng and Wieschaus, 2016*, *2017*). Our results suggest a similar mechanism since *Nvsnail* overexpression in endomesodermal lineages (*Figure 6—figure supplement 1*) is not sufficient to segregate cells and the endomesoderm remains as an epithelium. However, unlike *Drosophila*, Par proteins and AJs are not enhanced but degraded during the gastrulation of *N. vectensis* (*Figure 1*). As it is discussed in (*Weng and Wieschaus, 2017*), not only the degradation but also the turnover of AJs and Par proteins in adjacent epithelia is essential for EMT-mediated germ layer segregation in different animals. The dual identity of *N. vectensis* endomesoderm is characterized by the continuous expression of *Nvsnail* genes (*Martindale et al., 2004*) that repress the turnover of AJs and may play a role in inhibiting EMT from occurring (*Figures 6* and *7*).

Interestingly, components of the Wnt/PCP pathway are expressed only in the endomesoderm (*Kumburegama et al., 2011*; *Wijesena et al., 2011*), while components of the Par system are expressed only in the ectoderm (*Salinas-Saavedra et al., 2015*). It could be that NvSnail degrades AJs and inhibits their re-assembly in the endomesoderm, but the activation of contractile myosin by the Wnt/PCP pathway maintains the endomesodermal cells together in *N. vectensis* embryos. Hence in bilaterians, a mechanism (most likely downstream of Snail) that connects the cytoskeleton with cell-polarity may have evolved to tighten cell-cell adhesion in the endoderm and allow EMT.

To elucidate this, further comparative research and funding are needed to understand the cellular mechanisms that evolve to segregate mesoderm and control epithelial cell polarity at the base of the metazoan tree. For example, cnidarians, poriferans, and ctenophores present intriguing characteristics to study. In cnidarians, different modes of gastrulation have been described between species including unipolar and multipolar cell ingression and delamination (*Kirillova et al., 2018*; *Byrum and Martindale, 2004*; *Marlow and Martindale, 2007*). Poriferans display EMT-like

processes and cell morphologies during regeneration and trans-differentiation (*Nakanishi et al., 2014*; *Coutinho et al., 2017*). However, whether or not these processes involve similar molecular and cellular mechanisms are still unclear. Interestingly, ctenophores segregate a mesodermal cell population during embryogenesis but do not have the genes that encode for all cell-cell adhesion complexes and specify for mesoderm in bilaterians (*Figure 8—figure supplement 1*) (*Ganot et al., 2015*; *Ryan et al., 2013*). Thus, there is much to be learned by the comparative study of cell biology to understand the evolutionary origins of EMT and germ layer formation.

# Materials and methods

## Key resources table

| Reagent type (species) or resource | Designation | Source or reference | Identifiers | Additional information |
|---|---|---|---|---|
| Antibody | Mouse Anti-alpha-Tubulin Monoclonal Antibody, Unconjugated, Clone DM1A | Sigma-Aldrich | T9026; RRID:AB_477593 | (1:500) |
| Antibody | Anti-beta-Catenin antibody produced in rabbit | Sigma-Aldrich | C2206; RRID:AB_476831 | (1:300) |
| Antibody | Anti-Histone, H1 + core proteins, clone F152.C25.WJJ antibody | Millipore | MABE71; RRID:AB_10845941 | (1:300) |
| Antibody | anti-NvaPKC custom peptide antibody produced in rabbit | Bethyl labs; *Salinas-Saavedra et al. (2015)* | | Stored at MQ Martindale's lab; (1:100) |
| Antibody | anti-NvLgl custom peptide antibody produced in rabbit | Bethyl labs; *Salinas-Saavedra et al. (2015)* | | Stored at MQ Martindale's lab; (1:100) |
| Antibody | anti-NvPar-1 custom peptide antibody produced in rabbit | Bethyl labs; *Salinas-Saavedra et al. (2015)* | | Stored at MQ Martindale's lab; (1:100) |
| Antibody | anti-NvPar-6 custom peptide antibody produced in rabbit | Bethyl labs; *Salinas-Saavedra et al. (2015)* | | Stored at MQ Martindale's lab; (1:100) |
| Antibody | Goat anti-Mouse IgG Secondary Antibody, Alexa Fluor 568 | Thermo Fisher Scientific | A-11004; RRID:AB_2534072 | (1:250) |
| Antibody | Goat anti-Rabbit IgG Secondary Antibody, Alexa Fluor 647 | Thermo Fisher Scientific | A-21245; RRID:AB_2535813 | (1:250) |
| Antibody | Sheep Anti-Digoxigenin Fab fragments Antibody, AP Conjugated, Roche | Roche | 11093274910; RRID:AB_514497 | |
| Other | Alexa Fluor 488 Phalloidin | Thermo Fisher Scientific | A12379; RRID:AB_2315147 | (1:200) |
| Other | Texas Red Streptavidin | Vector Laboratories | SA-5006, RRID:AB_2336754 | (1:200) |
| Other | DAPI (4',6-Diamidino-2-Phenylindole, Dihydrochloride) | Thermo Fisher Scientific | D1306; RRID:AB_2629482 | (0.1 μg/μl) |
| Chemical compound, drug | Protein kinase Cζ pseudosubstrate, myristoyl trifluoroacetate salt | Sigma-Aldrich | P1614 | |
| Chemical compound, drug | 1-Azakenpaullone | Sigma-Aldrich | A3734 | |
| Chemical compound, drug | Biotinylated Dextran Amine-Texas Red | Vector Laboratories | SP-1140; RRID:AB_2336249 | |

*Continued on next page*

Continued

| Reagent type (species) or resource | Designation | Source or reference | Identifiers | Additional information |
|---|---|---|---|---|
| Chemical compound, drug | Dextran, Alexa Fluor 488; 10,000 MW, Anionic, Fixable | Thermo Fisher Scientific | D22910 | |
| Chemical compound, drug | Dextran, Alexa Fluor 555; 10,000 MW, Anionic, Fixable | Thermo Fisher Scientific | D34679 | |
| Chemical compound, drug | Dextran, Alexa Fluor 647; 10,000 MW, Anionic, Fixable | Thermo Fisher Scientific | D22914 | |
| Chemical compound, drug | Dextran, Cascade Blue, 10,000 MW, Anionic, Lysine Fixable | Thermo Fisher Scientific | D1976 | |
| Deposited Data | *Nematostella vectensis* genome assembly 1.0 | JGI | https://genome.jgi.doe.gov/Nemve1/Nemve1.home.html | |
| Biological sample | *Nematostella vectensis* | Whitney Laboratory for Marine Bioscience, FL, USA. | RRID:SCR_005153 | |
| Sequence-based reagent | Genomic PCR *Nvpar-6* and dn*Nvpar-6*: F-AAAACCAC CATCAGCCGAGTCA; R-TATTGATAGAATACCAGTCTCA | | NEMVEDRAFT _v1g233358 | |
| Sequence-based reagent | sgRNAs *Nvpar-6*: 1-GGATGTTGCCGACTCGCAGT; 2-GGAGAAGGCGAACTCGTCTG; 3-GGATAACCCTGTGCCAGTCA | CRISPRevolution sgRNA; Synthego | NEMVEDRAFT _v1g233358 | |
| Sequence-based reagent | dn*Nvpar-3*: F-ATGATGAAGGTTGTAGT; R-TGCGCCCGATTCGAATCCATCT | | NEMVEDRAFT _v1g240248 | |
| Sequence-based reagent | sgRNAs *Nvpar-3*: 1-GGGTGTTCGAGGGACGCGAT; 2- GGGCAGGTTTATCCCGAAGG; 3- ACCAACGAUCUAGAUCCAGU | CRISPRevolution sgRNA; Synthego | NEMVEDRAFT _v1g240248 | |
| Sequence-based reagent | Genomic PCR *Nvpar-3*: F-GTAGACGGGACTGGTTTGGA; R-AGGGACAGGTTGCTCCTTTT | | NEMVEDRAFT _v1g240248 | |
| Sequence-based reagent | dn*Nvpar-1*: F-AATATAAACTATGAACTTAACG; R-TTAAAGTTTTAATTCATTTGCA | | NEMVEDRAFT _v1g 139527 | |
| Sequence-based reagent | sgRNAs *Nvsnail-A*: 2-GGGGCCGGTAATGACGCGCG; 3-GGCGTAGAGTCACACCGCAA; 4-GGCGATGATATCGAGCTCGG; 5-GGGCATCTTGAGTGCACCCA | CRISPRevolution sgRNA; Synthego | NEMVEDRAFT _v1g240686 | |
| Sequence-based reagent | sgRNAs *Nvsnail-A*: 1-GGGCTCTCTTGCTCCGTAAC; 6-GGGTTTCCTGGCGCTGGGAT | CRISPRevolution Modified sgRNA; Synthego | NEMVEDRAFT _v1g240686 | |
| Sequence-based reagent | Genomic PCR *Nvsnail-A* (full length): F-ATGCCCCGCTCGTTTCTAG; R-TCCTTGTGACGGGCAGCC | | NEMVEDRAFT _v1g240686 | |
| Sequence-based reagent | sgRNAs *Nvsnail-B*: 1-GGAAGAGGATGTGAGGTTTT; 2-GAGATGATATTAGGCTGGTG; 4-GAAAAGCTGTACGACTCCTT; 5-GGGCATCTTGAGAGCGCCCA | CRISPRevolution sgRNA; Synthego | NEMVEDRAFT _v1g236363 | |
| Sequence-based reagent | sgRNAs *Nvsnail-B*: 3-GGGTGAAGACTAAGACAGAG; 6-GGGCGATGAATCGTGTTTAA | CRISPRevolution Modified sgRNA; Synthego | NEMVEDRAFT _v1g236363 | |

*Continued on next page*

*Continued*

| Reagent type (species) or resource | Designation | Source or reference | Identifiers | Additional information |
|---|---|---|---|---|
| Sequence-based reagent | Genomic PCR *Nvsnail-B* (full length): F-ATGCCGAGGTCCTTCCTGG; R-GCAGAGATTTTGCCGACACAT | | NEMVEDRAFT _v1g236363 | |
| Recombinant DNA reagent | pSPE3-mVenus | *Roure et al. (2007)* | | Gateway vector |
| Recombinant DNA reagent | pSPE3-mCherry | *Roure et al. (2007)* | | Gateway vector |
| Recombinant DNA reagent | pSPE3-*Nvpar-6*-mVenus | *Salinas-Saavedra et al. (2015)* | | |
| Recombinant DNA reagent | pSPE3-*Nvpar-3*-mVenus | *Salinas-Saavedra et al. (2015)* | | |
| Recombinant DNA reagent | *Nvß*-catenin expression constructs | *Wikramanayake et al. (2003)*; *Röttinger et al., 2012* | | |
| Recombinant DNA reagent | *Nvsnail-A* backbone to generate expression constructs | *Magie et al. (2007)* | | |
| Software, algorithm | Fiji (ImageJ) | NIH | http://fiji.sc | |
| Software, algorithm | Imaris 7.6.4 | Bitplane Inc. | | |
| Software, algorithm | CRISPRscan | *Moreno-Mateos et al. (2015)* | http://www.crisprscan.org/ | |
| Software, algorithm | SPSS | IBM | | |

## Culture and spawning of *Nematostella vectensis*

Spawning, gamete preparation, fertilization and embryo culturing of *N. vectensis* (RRID:SCR_005153) embryos was performed as previously described (*Röttinger et al., 2012*; *Hand and Uhlinger, 1992*; *Layden et al., 2013*; *Wolenski et al., 2013*). Adult *N. vectensis* were cultivated at the Whitney Laboratory for Marine Bioscience of the University of Florida (USA). Males and females were kept in separate glass bowls (250 ml) in 1/3x seawater (salinity: 12pp) reared in dark at 16°C. Animals were fed freshly hatched *Artemia* three times a week and macerated oyster the day before spawning. Spawning was induced by incubating the adults under an eight-hour light cycle at 25°C the night before the experiment. Distinct groups of animals were spawned once every 2 weeks. Oocytes and sperm were collected separately and fertilized in vitro by adding sperm to egg masses for 25 min. The jelly mass surrounding the fertilized eggs was removed by incubating the eggs in 4% L-Cysteine (in 1/3x seawater; pH 7.4) for 15–17 min and then washed 3 times with 1/3x seawater. De-jellied eggs were kept in glass dishes (to prevent sticking) in filtered 1/3 seawater at 16°C until the desired stage.

## Immunohistochemistry

All immunohistochemistry experiments were carried out using the previous protocol for *N. vectensis* (*Salinas-Saavedra et al., 2015*) with a slight modification in the glutaraldehyde concentration to allow better antibody penetration. Embryos were fixed on a rocking platform at room temperature in two consecutive steps. Embryos of different stages were fixed for no longer than 3 min in fresh Fix-1 (100 mM HEPES pH 6.9; 0.05M EGTA; 5 mM MgSO4; 200 mM NaCl; 1x PBS; 3.7% Formaldehyde; 0.2% Glutaraldehyde; 0.5% Triton X-100; and pure water). Then, Fix-1 was removed and replace with fresh Fix-2 (100 mM HEPES pH 6.9; 0.05M EGTA; 5 mM MgSO4; 200 mM NaCl; 1x PBS; 3.7% Formaldehyde; 0.05% Glutaraldehyde; 0.5% Triton X-100; and pure water). Embryos were incubated in Fix-2 for 1 hr. Fixed embryos were rinsed at least five times in PBT (PBS buffer plus 0.1% BSA and 0.2% Triton X-100) for a total period of 3 hr. PBT was replaced with 5% normal goat serum (NGS; diluted in PBT) and fixed embryos were blocked for 1 to 2 hr at room temperature with gentle rocking. Primary antibodies were diluted in 5% NGS to desired concentration. Blocking

solution was removed and replaced with primary antibodies diluted in NGS. All antibodies incubations were conducted over night on a rocker at 4°C. After incubation of the primary antibodies, samples were washed at least five times with PBT for a total period of 3 hr. Secondary antibodies were then applied (1:250 in 5% NGS) and samples were left on a rocker overnight at 4°C. Samples were then washed with PBT and left on a rocker at room temperature for an hour. To visualize F-actin, samples were incubated then for 1.5 hr in Phalloidin (Invitrogen, Inc. Cat. # A12379) diluted 1:200 in PBT. Samples were then washed once with PBT and incubated with DAPI (0.1 µg/µl in PBT; Invitrogen, Inc. Cat. # D1306) for 1 hr to allow nuclear visualization. Stained samples were rinsed again in PBS two times and dehydrated quickly into isopropanol using the gradient 50, 75, 90, and 100%, and then mounted in Murray's mounting media (MMM; 1:2 benzyl benzoate:benzyl alcohol) for visualization. Note that MMM may wash DAPI out of your sample. For single blastomere microinjection experiments, after Phalloidin staining, samples were incubated with Texas Red Streptavidin (1:200 in PBT from 1 mg/ml stock solution; Vector labs, Inc. Cat.# SA-5006. RRID:AB_2336754) for 1 hr to visualize the injected dextran. We scored more than 1000 embryos per each antibody staining and confocal imaged more than 50 embryos at each stage.

The primary antibodies and concentrations used were: mouse anti-alpha tubulin (1:500; Sigma-Aldrich, Inc. Cat.# T9026. RRID:AB_477593), rabbit anti-ß-catenin (1:300; Sigma-Aldrich, Inc. Cat.# C2206. RRID:AB_476831), mouse anti-histone H1 (1:300; F152.C25.WJJ, Millipore, Inc. RRID:AB_10845941).

Rabbit anti-*Nv*aPKC, rabbit anti-*Nv*Lgl, rabbit anti-*Nv*Par-1, and rabbit anti-*Nv*Par-6 antibodies are custom made high affinity-purified peptide antibodies that were previously raised by the same company (Bethyl Inc.). All these four antibodies are specific to *N. vectensis* proteins (*Salinas-Saavedra et al., 2015*) and were diluted 1:100.

Secondary antibodies are listed in Key resources table.

## Fluorescent tracer dye penetration assay

Primary polyps were incubated and mounted in 1/3 sea water with fluorescent dextran solution (0.5 mg/ml). For uninjected embryos we used Dextran, Alexa Fluor 555 (Molecular Probes, INC. Cat.# D34679). For injected embryos, expressing fluorescent proteins, we used Dextran, Alexa Fluor 647 (Molecular Probes, INC. Cat.# D22914). Animals were observed within 10 min of incubation. 15 animals were recorded per treatment. For better visualization of the dextran solution inside the gastric cavity as shown in *Figure 2B*, we delivered additional dextran solution by microinjecting dye through the polyp's mouth. For the rest of the experiments, we mainly focused in the ectodermal permeability and we let the polyps to eat the solution by themselves as grown babies.

## mRNA microinjections

The coding region for each gene of interest was PCR-amplified and cloned into pSPE3-mVenus or pSPE3-mCherry using the Gateway system (*Roure et al., 2007*). Eggs were injected directly after fertilization as previously described (*Salinas-Saavedra et al., 2015*; *Layden et al., 2013*; *DuBuc et al., 2014*) with the mRNA encoding one or more proteins fused in frame with reporter fluorescent protein (N-terminal tag) using final concentrations of 450 ng/µl for each gene. Fluorescent dextran was also co-injected to visualize the embryos. For single blastomere microinjections, we raised the embryos until 8–16 cell stages (3–4 hpf) and co-injected the mRNA solution with Biotinylated Dextran Amine-Texas Red (10 µg/µl; Vector labs, Inc. Cat.# SP-1140. RRID:AB_2336249). Live embryos were kept at 16°C and visualized after the mRNA of the FP was translated into protein (2–3 hr). To avoid lethality, lower mRNA concentrations of the mutant proteins (250 ng/µl) were used to image the specimens for *Figures 2* and *4*, and *Figure 3—video 1*. Live embryos were mounted in 1/3 sea water for visualization. Images were documented at different stages from 3 to 96 hr post fertilization. We injected and recorded more than 500 embryos for each injected protein and confocal imaged approximately 20 specimens for each stage for detailed analysis of phenotypes *in vivo*. We repeated each experiment at least five times obtaining similar results for each case. The fluorescent dextran and primers for the cloned genes are listed in Key resources table.

## CRISPR/Cas9 knock-outs

To target our gene of interest, we used synthetic guide RNAs (sgRNA; Synthego, Inc.) and followed the instructions obtained from the manufacturer to form the RNP complex with Cas9 (Cas9 plus sgRNAs). Target sites, off-target sites, and CFD scores were identified and sgRNA were designed using CRISPRscan (*Doench et al., 2014*; *Moreno-Mateos et al., 2015*). We delivered the RNP complex by microinjection as previously described (*Servetnick et al., 2017*; *Wijesena et al., 2017*; *Ikmi et al., 2014*. Lyophilized Cas9 (PNA Bio., Inc. Cat.# CP01) was reconstituted in nuclease-free water with 20% glycerol to a final concentration of 2 µg/µl. Reconstituted Cas9 was aliquoted for single use and stored at −80°C. Embryos were injected, as described for mRNA microinjections, with a mixture (12.5 µl) containing sgRNAs (80 ng/µl of each sgRNA), Cas9 (3 µg), and Alexa Fluor 488-dextran (0.2 µg/µl; Molecular Probes, Inc. Cat.# D22910). Cas9 and sgRNA guides only controls were injected alongside each round of experiments. sgRNA guides controls are only shown in figures as Cas9 had no significative effects. 3 sgRNA were used to knock out *Nvpar-3*, 3 sgRNA were used to knock out *Nvpar-6*, 6 sgRNA were used to knock out *Nvsnail-A*, and 6 sgRNA were used to knock out *Nvsnail-B*. Single-embryo genomic DNA was analyzed as previously described (*Servetnick et al., 2017*). Gene expression was confirmed by in situ hybridization. We injected and recorded more than 1000 embryos for each treatment. We repeated each experiment at least six times obtaining similar results for each case. sgRNAs' sequences and PCR primers flanking the targeted region are listed in Key resources table.

## In situ hybridization

In situ hybridization was carried out following a previously published protocol for *N. vectensis* (*Wolenski et al., 2013*). Animals were fixed in ice-cold 4% paraformaldehyde with 0.2% glutaraldehyde in 1/3x seawater for 2 min, followed by 4% paraformaldehyde in PBTw for 1 hr at 4°C. Digoxigenin (DIG)-labeled probes, previously described (*Salinas-Saavedra et al., 2015*; *Röttinger et al., 2012*), were hybridized at 63°C for 2 days and developed with the enzymatic reaction of NBT/BCIP as substrate for the alkaline phosphatase conjugated anti-DIG antibody (Roche, Inc. Cat. #11093274910. RRID:AB_514497). Samples were developed until gene expression was visible as a purple precipitate.

## Drug treatment

We incubated *N. vectensis* embryos in 20 µM of aPKC pseudosubstrate inhibitor (Protein kinase Cζ pseudosubstrate, myristoyl trifluoroacetate salt, Sigma, Cat.#P1614) from 0 to 4 hpf. Controls and 1-azakenpaullone (AZ; Sigma, Cat.#A3734) drug treatment of *N. vectensis* embryos was performed as previously described (*Röttinger et al., 2012*; *Leclère et al., 2016*). Embryos were developed in 5 µm AZ from 3 to 76 hpf. Controls were incubated in 0.08% DMSO.

## Imaging of *N. vectensis* embryos

Images of live and fixed embryos were taken using a confocal Zeiss LSM 710 microscope using a Zeiss C-Apochromat 40x water immersion objective (N.A. 1.20). Pinhole settings varied between 1.2 and 1.4 A.U. according to the experiment. The same settings were used for each individual experiment to compare control and experimental conditions. Results from in situ hybridization studies were imaged using a Zeiss Imager.M2 with a Zeiss 425 HRc color digital camera run by Zeiss Zen 2012 software. Z-stack images were processed using Imaris 7.6.4 (Bitplane Inc.) software for three-dimensional reconstructions and FIJI for single slice and movies. Final figures were assembled using Adobe Illustrator and Adobe Photoshop.

## Co-Immunoprecipitation

Tissue homogenization and protein extraction was performed as described in (*Salinas-Saavedra et al., 2015*; *Suzuki et al., 2001*; *Wang et al., 2012*). Briefly, embryos were homogenized in 200 µl of ice cold lysis buffer (30 mM HEPES, pH to 7.5, 1 mM EDTA, 150 mM NaCl, 50 mM NaF, 1 mM Na3VO4, 1 mM Na2MoO4, 1 mM MgCl2, 1% NP-40, 10% Glycerol, Protease Inhibitor cocktail (Sigma P8340) and PMSF. After 15 min' incubation on ice, crude lysate was carefully laid on top of a 200 µl sucrose cushion (1M sucrose, 30 mM HEPES, pH to 7.5, 1 mM EDTA, 150 mM NaCl, 50 mM NaF, 1 mM Na3VO4, 1 mM Na2MoO4) and yolk pelleted by centrifugation at 1000 rpm for 10 min.

The top layer was transferred to a clean microcentrifuge tube and 300 µl of lysis buffer was added. Approximately, 5 mg of protein was obtained from 60 µl of embryos (more than 15,000 embryos) homogenized in 500 µl of lysis buffer. 2 mg of total protein (in 500 µl of lysate buffer) was incubated with Par-specific antibodies cross-linked to Pierce Protein A/G Magnetic Beads (Pierce Biotechnology, Rockford, IL). (Pre-immune) IgG-IP pull downs were performed as a negative control for each experiment. Three antibodies, described previously (*Salinas-Saavedra et al., 2015*), against three different proteins (NvaPKC, NvPar-6, and NvPar-1) were utilized. We performed co-IP experiments using early cleavage (2–4 hpf) and for gastrula stages (24–30 hpf) lysates. Co-IP experiments were repeated four times for each stage using fresh lysates every time. For a detailed protocol, please, go to http://www.whitney.ufl.edu/research/faculty/mark-q-martindale/mark-q-martindale-lab-protocols/

## Morphometric measurements

Epithelial thickness was measured using confocal images of embryos immunohistochemically labeled, processed with Imaris 7.6.4 (Bitplane Inc.). For detailed and graphical explanation, please see *Figure 3—figure supplement 3* and *Figure 3—figure supplement 7*. For each treatment, epithelial thickness was determined by the average cell length (µm) along the apico-basal axis (A-B axis) of five individual cells. These values were made for two perpendicular axes Axis1 (A1) and Axis2 (A2) and normalized by the embryonic diameter (µm), in order to minimize technical artifacts (e.g. fixation and mounting) that could have affected the shape/size of the cell. Giving a proportion (p) that was calculated for A1 and A2, named p1 and p2, respectively. The values of p1 and p2 obtained for 90 control embryos, were statistically compared with the respective values obtained for 103 mutant-embryos using the Mann–Whitney U test (nonparametric; normality was tested using SPSS software) with a critical p-value of 0.05. The null hypothesis assumed no differences between the cell sizes of control and mutant-embryos. Values and statistics can be found in *Figure 3—figure supplement 3—source data 1*.

## Acknowledgements

We thank Leslie S Babonis for TEM data in *Figure 1—figure supplement 1*, A Wikramanayake for *Nvß*-catenin expression constructs, and C Magie and E Röttinger for *Nvsnail-A* expression constructs. We thank E Seaver, CE Schnitzler, and the members of Martindale's lab for helpful discussion. We also thank the *NSF-IOS 1239422- Broadening participation of underrepresented groups in Developmental Biology* and NSF REU (DBI-1156528) programs for AQR.

## Additional information

### Funding

| Funder | Grant reference number | Author |
| --- | --- | --- |
| National Institutes of Health | GM093116 | Mark Q Martindale |
| National Science Foundation | IOS-1755364 | Mark Q Martindale |
| National Aeronautics and Space Administration | 16-EXO16_2-0041 | Mark Q Martindale |
| Synthego | Grants | Mark Q Martindale |

The funders had no role in study design, data collection, and interpretation, or the decision to submit the work for publication.

### Author contributions

Miguel Salinas-Saavedra, Conceptualization, Supervision, Funding acquisition, Validation, Investigation, Visualization, Methodology, Writing—original draft, Project administration, Writing—review and editing; Amber Q Rock, Validation, Investigation, Writing—review and editing; Mark Q Martindale, Conceptualization, Resources, Supervision, Funding acquisition, Validation, Investigation, Visualization, Methodology, Writing—original draft, Project administration, Writing—review and editing

## Author ORCIDs
Miguel Salinas-Saavedra (iD) http://orcid.org/0000-0002-1598-9881
Mark Q Martindale (iD) https://orcid.org/0000-0002-5805-5640

## Decision letter and Author response
Decision letter https://doi.org/10.7554/eLife.36740.044
Author response https://doi.org/10.7554/eLife.36740.045

## Additional files

### Supplementary files
• Transparent reporting form
DOI: https://doi.org/10.7554/eLife.36740.040

### Data availability

No major datasets were generated in this study. All data generated or analyzed during this study are included in the manuscript and supporting files. All raw images are available on request.

The following previously published dataset was used:

| Author(s) | Year | Dataset title | Dataset URL | Database, license, and accessibility information |
|---|---|---|---|---|
| Putnam NH, Srivastava M, Hellsten U, Dirks B, Chapman J, Salamov A, Terry A, Shapiro H, Lindquist E, Kapitonov VV, Jurka J, Genikhovich G, Grigoriev IV, Lucas SM, Steele RE, Finnerty JR, Technau U, Martindale MQ, Rokhsar DS | 2007 | *Nematostella vectensis* genome assembly JGI 1.0 | http://genome.jgi.doe.gov/Nemve1/Nemve1.home.html | Publicly available at the JGI Genome Portal |

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
