## [Decision Letter]

Thank you for submitting your article "Germ layer specific regulation of cell
polarity and adhesion gives insight into the evolution of mesoderm." for
consideration by *eLife*. Your article has been reviewed by two peer
reviewers, and the evaluation has been overseen by Alejandro Sánchez Alvarado as the
Reviewing Editor and Patricia Wittkopp as the Senior Editor. The following
individual involved in review of your submission has agreed to reveal his identity:
David Matus (Reviewer #2).

The reviewers have discussed the reviews with one another and the Reviewing Editor
has drafted this decision to help you prepare a revised submission.

Summary:

Salinas-Saavedra et al., build a cell biological argument for a plausible
evolutionary scenario of "diploblastic" endomesoderm giving rise to
separate endoderm and mesoderm of the bilaterians. Using functional studies, the
authors show that Nematostella has the molecular tools to segregate a mesoderm, but
does not use it in this context. They disrupted the apicobasal cell-polarity that is
mediated by the aPKC/Par complex, resulting in mesenchymal-like cells that do not
form naturally in gastrulating anemones; hence, in a way, the authors generated the
first stages of an artificial 'mesoderm'. A key finding supporting the proposed
scenario is that disruption of Par-3 or mis-expression of the transcription factors
SnailA/B induce EMT-like behavior from ectodermal cells. The implications from these
results are that (1) Nematostella cells have maintained the capacity to undergo EMT
even though they may have lost EMT cell behaviors earlier in cnidarian evolution; or
that (2) EMT is easily "re-wirable" as long as the correct cellular
components are in place. Altogether, the findings reported suggest a scenario
according to which an ancient EMT-like mechanism, present in common
cnidarian-bilaterian ancestor, was co-opted for mesoderm segregation in the stem
Bilateria. Given the diversity of gastrulation mechanisms in Cnidaria, examining
whether other cnidarians utilize EMT during gastrulation should be informative. As
EMT behaviors have been described descriptively during regeneration and
trans-differentiation in sponges (see Coutinho et al., 2017) and Nakanishi et al.,
2014), it is plausible that other Cnidaria may utilize EMT during gastrulation, or
that at EMT may have evolved prior to the Cnidarian-Bilaterian divergence instead.
The implications of this work are not only relevant to our understanding of the
evolution of mesoderm, but to cell biologists interested in the evolution and
plasticity of EMT, which is a critical cell behavior during development and disease
states such as cancer dissemination.

Essential revisions:

1) In subsection “The *Nv*aPKC/Par complex regulates the formation and
maintenance of cell-cell junctions” the authors write: "dnNvPar-1 can be
phosphorylated by aPKC but cannot phosphorylate the aPKC/Par complex (Böhm,
Brinkmann, Drab, Henske, and Kurzchalia, 1997; Vaccari, Rabouille, and Ephrussi,
2005). Thus, dnNvPar-1 can localize to the cell cortex where aPKC is inactive."
However, the authors do not have any evidence for phosphorylation/kinase activity
for Nematostella dnPar-1. The authors should qualify this statement or provide
experimental evidence.

2) In subsection “The *Nv*aPKC/Par complex regulates the formation and
maintenance of cell-cell junctions”, the authors conclude that Par3 KO has no
significant effect on lineage markers; however, Figure 3—figure supplement 1E is not
quite consistent with this notion, in particular since the percentages of KO embryos
shown on the PCR gel on Figure 2—figure supplement 1 is low (2/12); it is of course
possible that animals with a wild type size band (i.e., about 985 bp) are also
mutated (e.g. small deletion resulting in a frame shift), but this isn't known since
the authors didn't sequence the PCR products. Hence, the embryos with defective
lineage marker expression could be the mutated ones.

3) Figure 3—figure supplement2A – The change in epithelial integrity of the ectoderm
following Par6 and Par3 disruption is quite stunning. It would be important for the
authors to quantify the range of thickness by measuring as compared to control
embryos and then use the appropriate statistical test to show that this is a
significant result. A dot plot may be the most effective way to display the range of
phenotypes observed in thickness of the epithelium. In other words, measuring
thickness of the epithelium in a statistical framework. will benefit the work
greatly.

4) Figure 3—figure supplement 1; what do the thicker and thinner than normal
epithelia reflect? Is it just being abnormal, or is there a different interpretation
for thick vs thin?

5) In Figure 5—figure supplement 2C, the authors claim that no expression, or mosaic
expression, of SnailA+B occurred in SnailA+B KO embryos. However, they only show the
representative results of 75% of SnailA, and 49% of SnailB; how did the others look
like? Also, there are faint bands on the gel running the PCR products from the KO
embryos on panels A & B, suggesting that KO wasn't complete. It is not clear to
us how the authors concluded that ectodermal cell fate had not changed.

6) For review and visualization purposes, the figures are so small that it is
challenging to see some of the most striking results at cellular resolution. Please
provide a representative z-stack for data from some of the experiments. For example,
Figure 5D would benefit from doing this, to better see bottle cell morphology
following Snail induction.

---

## [Author Response]

[…] Given the diversity of gastrulation mechanisms in Cnidaria, examining whether
other cnidarians utilize EMT during gastrulation should be informative. As EMT
behaviors have been described descriptively during regeneration and
trans-differentiation in sponges (see Coutinho et al., 2017) and Nakanishi et
al., 2014), it is plausible that other Cnidaria may utilize EMT during
gastrulation, or that at EMT may have evolved prior to the Cnidarian-Bilaterian
divergence instead. The implications of this work are not only relevant to our
understanding of the evolution of mesoderm, but to cell biologists interested in
the evolution and plasticity of EMT, which is a critical cell behavior during
development and disease states such as cancer dissemination.

We completely agree with this comment and further address this issue in the
Conclusion section.

Essential revisions:1) in subsection “The NvaPKC/Par complex regulates the formation and maintenance
of cell-cell junctions” the authors write: "dnNvPar-1 can be phosphorylated
by aPKC but cannot phosphorylate the aPKC/Par complex (Böhm, Brinkmann, Drab,
Henske, and Kurzchalia, 1997; Vaccari, Rabouille, and Ephrussi, 2005). Thus,
dnNvPar-1 can localize to the cell cortex where aPKC is inactive." However,
the authors do not have any evidence for phosphorylation/kinase activity for
Nematostella dnPar-1. The authors should qualify this statement or provide
experimental evidence.

The reviewers are correct. We have added an additional figure where we show co-IP
experiments using the NvPar-1 specific antibody (Figure 3—figure supplement 2)
showing that NvPar-1 interacts with NvaPKC and NvPar-6 (both proteins detected with
their own specific antibodies). In addition, we observed three bands labeled with
NvPar-1 antibody around 80 KD, suggesting different phosphorylation states of this
protein, which may be a product of NvaPKC activity.

However, we were not able to perform co-IP experiments on the dominant negative form
of NvPar-1 because of technical difficulties. To perform the co-IP, we needed to
extract protein from over 13,000 wild type embryos; we were not able to inject and
express the dominant negative form of this protein into this large number of
embryos. Hence, we changed our statement to:(subsection “The
*Nv*aPKC/Par complex regulates the formation and maintenance of
cell-cell junctions”) “Since NvPar-1 is phosphorylated by NvaPKC (Figure 3—figure
supplement 2), we predict that, as in other systems, dnNvPar-1 could be
phosphorylated by NvaPKC but would not phosphorylate the NvaPKC/Par complex [37,38].
Thus, dnNvPar-1 can localize to the cell cortex where aPKC may be inactive.”

2) In subsection “The NvaPKC/Par complex regulates the formation and maintenance
of cell-cell junctions”, the authors conclude that Par3 KO has no significant
effect on lineage markers; however, Figure 3—figure supplement 1E is not quite
consistent with this notion, in particular since the percentages of KO embryos
shown on the PCR gel on Figure 2—figure supplement 1 is low (2/12); it is of
course possible that animals with a wild type size band (i.e., about 985 bp) are
also mutated (e.g. small deletion resulting in a frame shift), but this isn't
known since the authors didn't sequence the PCR products. Hence, the embryos
with defective lineage marker expression could be the mutated ones.

As we show, the endomesoderm and the overall morphology of the embryos were
drastically affected by Par3 KO making it hard to interpret the phenotype. The
complete zygotic deletion of the proteins NvPar6 and NvPar3 resulted in lethal
phenotypes (embryos developed with maternal contributions during earlier stages) and
since all stable mutant embryos could have had some level of mosaicism where the
wild type gene/protein may have been present in their ectoderm, we chose to assess
our experiments by ISH and IHC using our specific probes, a specific antibody
against NvPar-6, and a ß-catenin antibody.

In order to make this clearer, we have rephrased this statement:

(subsection “The *Nv*aPKC/Par complex regulates the formation and
maintenance of cell-cell junctions”) “Although it was difficult to dissect
significant changes in the expression of germ layer markers (e.g. Nvbra, Nvsnail,
NvSix3/6, and Nvfz10) from the morphological changes associated with epithelial
integrity when these genes were disrupted (Figure 3—figure supplement 4E), it is
clear that the primary defect in NvPar3 KO were aspects of cell adhesion and not
cell type specification.”

3) Figure 3—figure supplement2A – The change in epithelial integrity of the
ectoderm following Par6 and Par3 disruption is quite stunning. It would be
important for the authors to quantify the range of thickness by measuring as
compared to control embryos and then use the appropriate statistical test to
show that this is a significant result. A dot plot may be the most effective way
to display the range of phenotypes observed in thickness of the epithelium. In
other words, measuring thickness of the epithelium in a statistical framework.
will benefit the work greatly.

Done, Figure 3—figure supplement 3.

4) Figure 3—figure supplement 1; what do the thicker and thinner than normal
epithelia reflect? Is it just being abnormal, or is there a different
interpretation for thick vs thin?

With the information that we have, we can only say that is an abnormality caused by
the loss of epithelial homeostasis. We have added a note to the figure legend in
Figure 3—figure supplement 3.

5) In Figure 5—figure supplement 2C, the authors claim that no expression, or
mosaic expression, of SnailA+B occurred in SnailA+B KO embryos. However, they
only show the representative results of 75% of SnailA, and 49% of SnailB; how
did the others look like?

The others look like the control expression. A note was added to the legend.

Also, there are faint bands on the gel running the PCR products from the KO
embryos on panels A & B, suggesting that KO wasn't complete.

As we show by ISH, there is some mosaicism present in the KO embryos. Similarly, to
the case for Par3 KO, we picked up a limited number of embryos out of over 2000
injected embryos, thus, the probability to have mosaicism or wild type embryos are
very high. We decided then to assess the phenotypes and results by IHC and ISH.

It is not clear to us how the authors concluded that ectodermal cell fate had not
changed.

We performed ISH for *brachyury, chordin*, and *six3/6*
that are well known ectodermal markers in *N. vectensis*. We did not
observe a significant change in gene expression even though the morphology of the
embryos was highly disrupted.

6) For review and visualization purposes, the figures are so small that it is
challenging to see some of the most striking results at cellular resolution.

We have separated some of the main figures (now 8 figures) and their supplemental
data in order to increase the size of important results. We are more than happy to
make any further modification as necessary.

Please provide a representative z-stack for data from some of the experiments.
For example, Figure 5D would benefit from doing this, to better see bottle cell
morphology following Snail induction.

We have included Videos and ‘*.gif’ files of the z-stacks for the most representative
results. We are more than happy to share more figures as necessary.

Z-stacks (*.gif’ format. It can be opened and edited using FIJI) were uploaded as
supplementary files:

Figure 3E z-stack: Related to Figure 3E CRISPR phenotype (Video 1).

Figure 4 z-stack NvPar-6: Related to Figure 4 CRISPR phenotype for NvPar-6
antibody.

Figure 4 z-stack ß-catenin: Related to Figure 4 CRISPR phenotype for ß-catenin
antibody (Video 2).

Figure 6D z-stack NvPar-6: Related to Figure 6D for NvPar-6 antibody (Video 5).

Figure 6D z-stack NvPar-1: Related to Figure 6D for NvPar-1 antibody (Video 6).